# A new form of diabetes caused by *INS* mutations defined by zygosity, stem cell and population data

Yue Tong [1✉], Marianne Becker [2], Ulrike Schierloh[2], Flávia Natividade da Silva[1], Leena Haataja[3], Ying Cai[1], Kashyap A Patel[4], Farah Kobaisi[5], Uyenlinh L Mirshahi [6], Kevin Colclough[7], Muhammad Shabab Javed [8], Matthew N Wakeling[4], Federica Fantuzzi[1], Maria Lytrivi[1,9], Toshiaki Sawatani[1], Maria Nicol Arroyo[1], Xiaoyan Yi[1], Chiara Vinci[1], Hossam Montaser [10], Nathalie Pachera[1], Timo Otonkoski [10,11], Mariana Igoillo-Esteve[1], Raphaël Scharfmann[5], Andrew T Hattersley [4], Peter Arvan [3], Carine De Beaufort[2] & Miriam Cnop [1,9,12✉]

## Abstract

The *INS* c.16 C > T (insulin p.Arg6Cys, R6C) variant was reported to cause autosomal dominant monogenic diabetes, yet its pathogenicity has been questioned. R6C preproinsulin exhibits impaired translocation into the endoplasmic reticulum (ER). We explored R6C pathogenicity using integrative clinical, genetic, and functional approaches. Homozygous *INS* R6C individuals presented early-onset insulin-treated diabetes, whereas heterozygous carriers showed variable or absent glycemic phenotypes. Population-level analysis revealed no significant enrichment of diabetes among heterozygotes. Heterozygous R6C patient's induced pluripotent stem cell (iPSC)-derived pancreatic β cells exhibited minimal defects, while homozygous R6C β cells displayed preproinsulin accumulation and reduced insulin content and secretion. In vivo, homozygous R6C β cell transplants recapitulated insulin deficiency and responded poorly to GLP-1 receptor agonist. Homozygous R6C β cells had a gene signature of attenuated translation, translocation and ER related pathways. Our findings establish R6C as a recessive loss-of-function mutation, prompting a clinical reassessment of heterozygous R6C carriers. This study highlights the power of population genetic databases, patients' iPSC-based modeling and multi-modal variant classification frameworks for dissecting the consequences of genetic variants in monogenic diabetes.

**Keywords** Monogenic Diabetes; *INS* R6C Variant; Rare Variant Penetrance; Population Genomics; iPSC-Derived β Cells
**Subject Categories** Genetics, Gene Therapy & Genetic Disease; Metabolism

## Introduction

Monogenic diabetes accounts for ~3% of diabetes cases diagnosed under 30 years and exhibits a broad spectrum of clinical presentations, including neonatal, maturity-onset diabetes of the young, and syndromic forms (American Diabetes Association Professional Practice, 2025; International Diabetes Federation, 2025). This proportion may be an underestimation, as it is often misdiagnosed as type 1 or type 2 diabetes. To date, mutations in over 70 genes have been described, most of which disrupt pancreatic β-cell development, function, and/or survival (Greeley et al, 2022; De Franco et al, 2023; Perera et al, 2023). Since the first description linking mutations in the *INS* gene (encoding insulin) to diabetes (Støy et al, 2007; Colombo et al, 2008), *INS* mutations have been recognized as a major cause of monogenic diabetes. While insulin plays a central role in glucose homeostasis, *INS* diabetes can manifest across a broad age range—from within the first 6 months of life to adulthood (Støy et al, 2021). Inheritance can be dominant or recessive, with penetrance varying substantially depending on the type and position of the mutation (Garin et al, 2010). This clinical heterogeneity underscores the need for careful interpretation of *INS* variants when encountered in diagnostic sequencing.

Depending on the affected domain or amino acid, *INS* mutations interfere with distinct stages of insulin biosynthesis, including mRNA translation into preproinsulin, signal peptide-mediated endoplasmic reticulum (ER) translocation, proinsulin folding and disulfide bond formation, and conversion into mature insulin. Most are heterozygous missense mutations affecting

[1]ULB Center for Diabetes Research, Université Libre de Bruxelles, Brussels, Belgium. [2]Pediatric Endocrinology and Diabetology (DECCP), Centre Hospitalier de Luxembourg, Luxembourg, Luxembourg. [3]Division of Metabolism, Endocrinology and Diabetes, University of Michigan Medical School, Ann Arbor, MI, USA. [4]Department of Clinical and Biomedical Sciences, Faculty of Health and Life Sciences, University of Exeter, Exeter, UK. [5]Université Paris Cité, Institut Cochin, INSERM U1016, CNRS UMR 8104, 75014 Paris, France. [6]Department of Genomic Health, Geisinger, Danville, PA, USA. [7]Exeter Genomics Laboratory, Royal Devon University Healthcare NHS Foundation Trust, Exeter, UK. [8]Department of Paediatrics, Walsall Healthcare NHS Trust, Walsall Manor Hospital, Walsall, UK. [9]Department of Endocrinology, ULB Erasmus Hospital, Brussels University Hospital, Université Libre de Bruxelles, Brussels, Belgium. [10]Stem Cells and Metabolism Research Program, Faculty of Medicine, University of Helsinki, Helsinki, Finland. [11]Children's Hospital, University of Helsinki and Helsinki University Hospital, Helsinki, Finland. [12]WEL Research Institute, Wavre, Belgium. ✉E-mail: yue.tong94@gmail.com; miriam.cnop@ulb.be

cysteine residues critical for disulfide bond formation and insulin folding, with highly penetrant dominant inheritance (Wang et al, 1999; Støy et al, 2007; Colombo et al, 2008; Wang et al, 2020). Unlike most monogenic diabetes genes, where haploinsufficiency underlies dominant inheritance, these dominant *INS* mutations act through a toxic gain-of-function mechanism where misfolded proinsulin induces severe β cell ER stress. Recessive *INS* mutations typically involve reduced insulin synthesis or production of nonfunctional and/or degradation-prone proteins, representing classic loss-of-function with variable penetrance (Støy et al, 2021; Garin et al, 2010; Tans et al, 2024).

Despite this growing mechanistic insight, many *INS* variants are reported only once and not functionally followed-up (Støy et al, 2007; Colombo et al, 2008; Molven et al, 2008; Boesgaard et al, 2010; Støy et al, 2021). The identification and interpretation of variants have been greatly facilitated by advances in genomic technologies and in silico predictive tools. However, assessing variants with subtle or context-dependent effects remains challenging (Wright et al, 2024). Large-scale and diverse population genetic databases such as gnomAD (Atkinson et al, 2023), TOPMed (Taliun et al, 2021), All of Us (All of Us Research Program Investigators et al, 2019), and UK Biobank (Sudlow et al, 2015) have revealed that certain rare variants—thought to be pathogenic based on limited family data—may be more prevalent than initially estimated.

Comprehensive clinical, genetic and cell studies remain essential for variant classification and to guide clinical care. Access to primary pancreatic islets from patients—especially those carrying rare mutations—is exceedingly rare if not impossible. Non-human models or human β cell lines may not fully recapitulate the patient-specific regulatory landscape and cellular context.

The NM_000207.3: c.16 C > T, p.(Arg6Cys) *INS* mutation (sixth arginine substituted by cysteine, henceforth referred to as R6C) exemplifies these challenges. It was first reported as an autosomal dominant mutation in a family with three heterozygous R6C individuals who were diagnosed with non-insulin-dependent, non-obese diabetes in adolescence or adulthood (Edghill et al, 2008). This mutation is located in the preproinsulin signal peptide. The signal peptide consists of three domains: a central hydrophobic h-region that binds with subunit SRP54 of the signal recognition particle (SRP) (Gutierrez Guarnizo et al, 2023), which targets nascent protein to the ER membrane, a positively charged n-region that promotes electrostatic interactions with the negatively charged phosphate group of lipids in the ER membrane ("positive-inside" rule (Nesmeyanova et al, 1997)), and a c-region containing the cleavage site recognized by signal peptidase. The n- and h-regions are critical for co-translational translocation of preproinsulin, guiding the ribosome–nascent chain complex to the SRP and enabling its translocation through the SEC61 translocon (Lang et al, 2017; Xu et al, 2024). Alterations in charge or hydrophobicity caused by R6C may potentially impair SRP recognition and ER targeting (Liu et al, 2015; Gutierrez Guarnizo et al, 2023; Miller et al, 2024; Sánchez et al, 2025).

*INS* R6C has been considered to cause autosomal dominant early-onset diabetes (Meur et al, 2010; Hussain et al, 2013; Støy et al, 2021; Chua et al, 2024), but later-onset cases have been described (Bansal et al, 2017). Autosomal dominant R6C diabetes could result from misfolded proinsulin, ER stress, and β-cell death (Støy et al, 2007; Liu et al, 2012; Balboa et al, 2018). However, R6C transfection in rat INS-1E β and HEK293T cells resulted in inefficient translocation of R6C preproinsulin across the ER membrane and 50% reduced insulin production, but no toxic gain-of-function (Guo et al, 2014, 2018). Time-resolved translocation assays support a primary ER targeting defect (Iwasa et al, 2025).

Here, we analyzed aggregated population data and challenged the assumption of autosomal dominant *INS* R6C inheritance, prompting reconsideration of its penetrance and disease mechanism. We took advantage of patients' iPSC-derived β cells, a species- and disease-specific model that replicates β cell failure seen in patients, to further the understanding of pathogenic mechanisms and bridge clinical genetics with β-cell biology.

## Results

### Homozygous R6C mutation leads to early-onset diabetes, while heterozygous carriers exhibit variable adult-onset hyperglycemia

The proband and her paternal uncle developed diabetes at age 11 and 9 years, with markedly elevated hemoglobin A1c (HbA1c ∼100 mM/M, ∼11%) and a BMI of 19.6 (percentile 79) and 20.8 kg/m$^2$ (Fig. 1A; Table 1, A-III-1 and A-II-2). The proband had a random C-peptide of 3.3 ng/mL (1092.3 pM) concomitant with a blood glucose level at 286 mg/dL (15.9 mM). In another family, the proband (Fig. 1B; Table 1, B-II-1) was diagnosed with diabetes at 13 years with HbA1c 78 mM/M (9.3%) and BMI 27.2 kg/m$^2$. All three were treated with insulin at diagnosis. Proband B-II-1 stopped insulin after 5 years due to excellent glycemic control (lowest HbA1c 39 mM/M, 5.7%). Two months later, her random C-peptide was 1.9 ng/mL (612 pM); 5 months later, her HbA1c was 49 mM/M (6.6%). Neither proband (A-III-1 and B-II-1) had ketoacidosis or β-cell autoantibodies GADA, IA2A, IAA, or ZnT8A at diagnosis or during follow-up. Genetic predisposition of type 1 diabetes was low (type 1 diabetes genetic risk score 15.2 and 21 centile of type 1 diabetes population, respectively (Oram et al, 2016)), suggestive of non-autoimmune diabetes. Targeted whole-genome sequencing revealed that proband A-III-1, 2 of her uncles (A-II-2 and A-II-1) and proband B-II-1 were homozygous carriers of the *INS* R6C variant. Overall genome-wide homozygosity in proband A-III-1 was low (1.7%), suggesting that the parents were not closely related. One region of homozygosity on chromosome 11 (chr11:206,682– 2,430,597) encompassed the insulin gene. The childhood to young adult onset of diabetes in homozygous R6C individuals, along with remaining endogenous insulin secretion, suggests a moderate or delayed impact on β cells.

We reviewed the clinical characteristics of heterozygous R6C individuals from this study (Fig. 1A,B) and the first reported family (Appendix Fig. S1A (Edghill et al, 2008)). Age at diabetes diagnosis was 36 ± 17 years and BMI 29 ± 5.2 kg/m$^2$ (mean ± SD). Most heterozygous individuals did not receive insulin treatment at diagnosis. The father of proband A-III-1 had a random C-peptide of 3.7 ng/mL (1225 pM) (Fig. 1A; Table 1, A-II-3) with concomitant blood glucose of 180 mg/dL (10 mM). Her mother had gestational diabetes at 27 and impaired glucose tolerance at 41 years (Fig. 1A; Table 1, A-II-4). The mother of proband B-II-1 (Fig. 1B; Table 1, B-I-2) had had gestational diabetes and was diagnosed with diabetes at 29 years. Another heterozygous same-site *INS* R6H mutation (NM_000207.3: c.17 G > A, p.(Arg6His)) has been suggested to result in milder β-cell dysfunction than heterozygous R6C (Appendix Fig. S1B (Guo et al, 2014, 2018; Meur et al, 2010)); these non-obese individuals had adult-onset diet-treated diabetes (Table 2). The clinical heterogeneity among heterozygous R6C and R6H individuals—ranging from impaired glucose tolerance to late-onset diabetes—raises questions about the penetrance of the variant and its classification as an

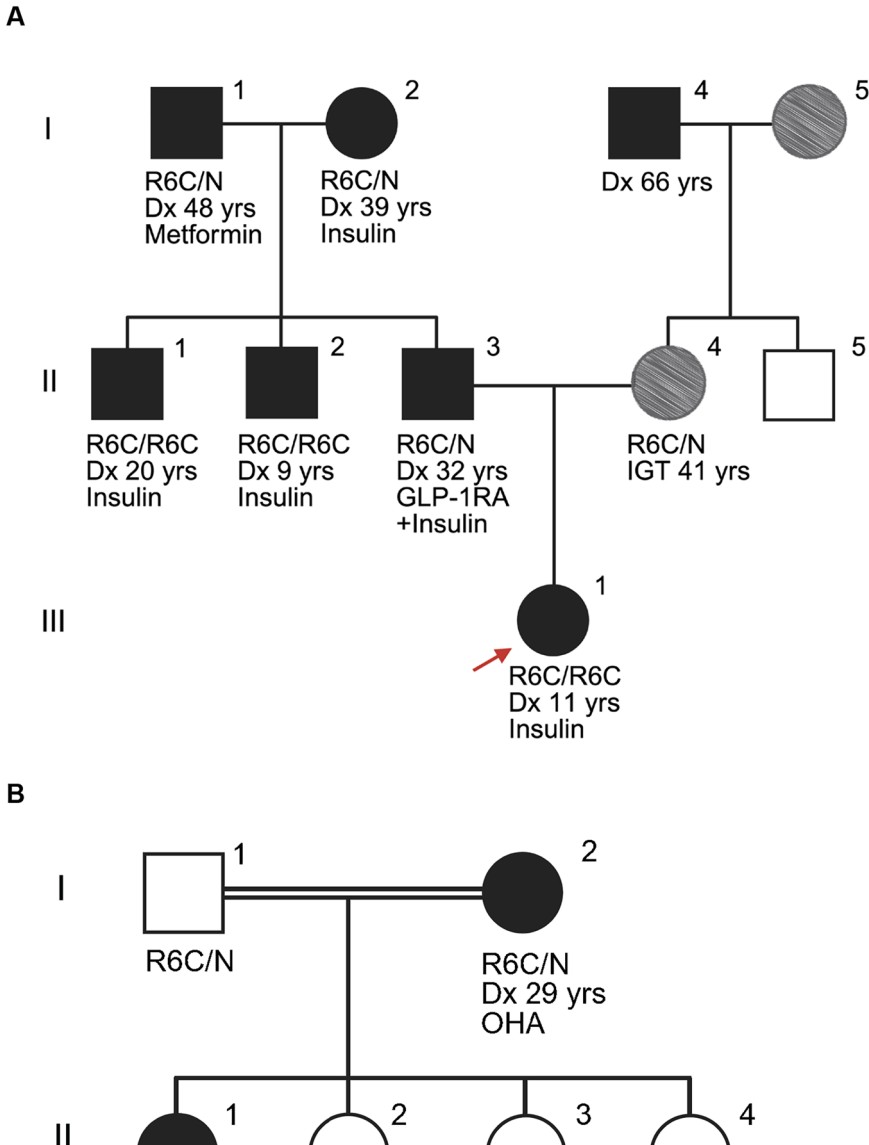

**Figure 1. Pedigree of the homozygous R6C families.**

Red arrow indicates the probands in the two families (A, B). Black solid symbol indicates diagnosis of diabetes, void symbol indicates unaffected individual, gray shaded indicates impaired glucose tolerance (IGT) or prediabetic. Roman numerals show generations and Arabic numerals individuals within each generation. R6C/R6C: homozygous, R6C/N: heterozygous. N/N non-carrier; other individuals lack genetic confirmation. Dx age at diagnosis, GLP-1RA glucagon-like peptide-1 receptor agonist, insulin injections or pump, OHA oral hypoglycemic agents.

autosomal dominant mutation. We therefore assessed its allele frequency in large-scale population databases.

## Integrative population, computational and structural evidence suggest limited pathogenicity of the heterozygous R6C variant

R6C allele frequencies were low although higher than usually seen for monogenic diabetes-causing *INS* variants, averaging around

0.00002, i.e. a heterozygote frequency of 1 in 25,000 individuals (Appendix Table S1) (Taliun et al, 2021; All of Us Research Program Investigators et al, 2019; Sun et al, 2024; ALFA (Allele Frequency Aggregator), 2024; Carey et al, 2016; Karczewski et al, 2020; Li et al, 2023b). All carriers in these large-scale databases were heterozygous, with notably higher frequencies observed in specific populations: 0.00016 in the Geisinger MyCode cohort, 0.0001 in the European population in the Allele Frequency Aggregator cohort, and 0.00033 in the Middle Eastern population

**Table 1. Clinical characteristics of *INS* R6C carriers.**

| *INS* mutation | R6C: c.16 C > T p.(Arg6Cys) | | | | | | | | | |
|---|---|---|---|---|---|---|---|---|---|---|
| Family | A | | | | | | | B | | |
| Identity | A-III-1 (proband) | A-II-1 | A-II-2 | A-II-3 | A-II-4 | A-I-1 | A-1-2 | B-I-1 | B-I-2 | B-II-1 (proband) |
| Homozygosity | yes | yes | yes | no | no | no | no | no | no | yes |
| Gender | female | male | male | male | female | male | female | male | female | female |
| **Clinical characteristics** | | | | | | | | | | |
| Age at diagnosis (years) | 11 | 20 | 9[a] | 32 | 41[b] | 48 | 39 | - | 29[c] | 13 |
| BMI (kg/m²) | 19.6 | 24.4 | 20.8 | 39 | - | - | - | 26.8 | 30.7 | 27.2 |
| HbA1c (%) | 11.9 | 11 | 11.4 | 10.2 | 6.2 | - | - | 5.4 | 6.5 | 9.3 |
| Treatment | Insulin | Insulin | Insulin | GLP-1RA + insulin | - | Metformin | Insulin | - | Metformin, gliclazide | Insulin[d] |

Variant described according to Human Genome Variation Society guidelines based on isoform NM_000207.3. Insulin was the first treatment for all homozygous patients.
[a]This individual stopped insulin treatment at 18 years on his own initiative, because of poorly controlled glycemia.
[b]The mother of the proband (A-II-4) was diagnosed with gestational diabetes and impaired glucose tolerance at age 41 years.
[c]The mother of the proband (B-I-2) was diagnosed with gestational diabetes.
[d]HbA1c was 7.2% 2 months after insulin initiation and 6.7% 10 months after. The proband reduced refined carbs, increased physical activity, and lost weight (BMI 22.7 kg/m²), with HbA1c 5.8% off treatment for nine months.

in gnomAD v4.1. There were four R6C heterozygotes in the UK Biobank, and none had diabetes. In the Geisinger cohort of 163,743 individuals, 51 heterozygous R6C carriers were identified (32 singletons and 18 from eight families), with no excess diabetes (odds ratio 1.4; 95% CI, 0.7–2.6). These findings challenge the assumption of a pathogenic effect for R6C heterozygosity; instead, it may be a context-dependent or low-risk variant with variable expressivity. The allele frequency of R6H is around 0.0001 to 0.0005 in gnomAD v4.1, i.e., a heterozygote carrier frequency of 1 in 5000 to 1000 people. In the UK Biobank, 6 of 86 R6H heterozygotes (7%) had diabetes, which was comparable to the general cohort prevalence (6.7%; 33,014 out of 490,029; Fisher's exact test, $P = 0.83$). Hence, both R6C and R6H is benign according to recent criteria (Karczewski et al, 2020; Huerta-Chagoya et al, 2024; Li and Polychronakos, 2024).

To further assess the potential impact of the R6C mutation, we employed a panel of in silico prediction tools using algorithms that integrate multiple sources of evidence, quested via dbNSFP version 5.1a (Liu et al, 2020). Three out of 6 classified R6C as damaging (Appendix Table S2). While these in silico predictions are conflicting, its location within the nascent preproinsulin signal peptide n-region might impair SRP recognition and ER SEC61 translocon orienting. The R6C mutation showed loss of positive charge in the n-region (Fig. EV1A) and increased hydrophobicity around the sixth residue (Fig. EV1B). We employed AlphaFold 3 (Abramson et al, 2024) to model the interaction between the preproinsulin signal peptide (amino acids 1–24) and SRP54. Compared to the wild-type signal peptide–SRP54 complex, the R6C complex showed a modest reduction in interaction quality and fewer hydrogen bonds (Fig. EV1C; Appendix Table S3). The SEC61α2, rather than SEC61α1, translocon has been shown to be crucial for insulin biosynthesis (Xu et al, 2024). AlphaFold 3 revealed an inverted orientation of the R6C signal peptide (Fig. EV1D; Appendix Table S3). This suggests weakened signal peptide–SRP54 interaction and misorientation of the signal peptide

**Table 2. Clinical characteristics of previously reported *INS* R6C and R6H carriers.**

| *INS* mutation | R6C: c.16 C > T p.(Arg6Cys) | | | R6H: c.17 G > A p.(Arg6His) | | |
|---|---|---|---|---|---|---|
| Family[d] | Edghill et al, 2008 | | | Meur G et al, 2010 | | |
| Identity | C-III-1[a] (proband) | C-II-2 | C-I-2 | D-II-3 (proband) | D-II-2 | D-III-2 |
| Homozygosity | no | no | no | no | no | no |
| Gender | female | female | female | male | male | female |
| **Clinical characteristics** | | | | | | |
| Age at diagnosis (years) | 15 | 15 | 65 | 20 | 51 | 26[c] |
| BMI (kg/m²) | 24.1 | 26.9 | 29.3 | 24.9 | 24.4 | 26.8 |
| HbA1c (%) | - | - | - | - | - | - |
| Treatment | OHA +Insulin[b] | OHA[b] | Diet[b] | Diet+OHA | Diet | Diet |

Variant described according to Human Genome Variation Society guidelines based on isoform NM_000207.3.
[a]In 2018, this proband was found to have a heterozygous *HNF1B* c.738 G > T p.(Leu246Phe) variant.
[b]Diet was the first treatment for C-III-1, C-II-2, C-I-2; oral hypoglycemic agents (OHA) and insulin were started later.
[c]The daughter of the proband's brother (D-III-1) was diagnosed with impaired glucose tolerance at 26 years and with gestational diabetes at 27 years.
[d]For the R6C individual with late-onset diabetes identified by Bansal et al, 2017 no clinical characteristics are available.

within the SEC61 translocon, which may compromise ER targeting and translocation efficiency of R6C insulin.

Building on the divergent clinical phenotypes in homozygous and heterozygous individuals (e.g., age at onset, BMI, treatment) and population-level allele frequencies, we next sought to determine to what extent the predicted structural alterations in SRP54 recognition and SEC61 orientation translate into pathological consequences in β cells.

## Complementation of the *INS* R6C mutant does not induce clonal β-cell death

We first asked whether expression of the *INS* R6C mutant in human β cells induces cell death. We generated a homozygous R6C model by introducing an insulin expression plasmid carrying the R6C variant into insulin-null EndoC-βH1 cells, with a wild-type plasmid as the control; a heterozygous model was generated by co-transfecting equal amounts of R6C and wild-type plasmids. Cell death was similar in wildtype and/or R6C mutant cells 1 and 3 days post transfection (Fig. EV2A; Appendix Fig. S2A,B). The transfection rate was relatively low, however (Appendix Fig. S2A,C). We therefore moved to patients' induced pluripotent stem cell (iPSC)-derived β cells, carrying the R6C allele at the endogenous *INS* locus, to model physiological expression levels and assess the mutation's impact in a patient-specific and disease-relevant context.

## Heterozygous R6C iPSC-derived β cells have moderate preproinsulin accumulation and non-affected proinsulin and insulin content and secretion

To understand whether heterozygous R6C is sufficient to disrupt the development, function and/or survival of pancreatic β cells, we generated heterozygous R6C patients' iPSCs from the proband's father (Fig. 1A; Table 1, A-II-3). We corrected the R6C variant using CRISPR/Cas9 technology, and we also obtained isogenic non-edited iPSCs (cells went through CRISPR but remained heterozygous R6C) (Appendix Fig. S3). All iPSC lines showed normal karyotype, expressed pluripotency markers, demonstrated capacity to spontaneously differentiate into three germ layers (Appendix Figs. S4, S5) and had no evidence of CRISPR-induced off-target effects (Appendix Table S3). Following an in vitro differentiation protocol (Fig. EV3A, top) that mimics embryonic development of β cells, both heterozygous R6C and corrected iPSCs differentiated normally into islets with comparable islet viability and similar proportions of insulin-expressing β-like cells (Figs. EV2B and EV3B). Heterozygous and corrected R6C cells followed a normal developmental pathway along the differentiation (Fig. EV3C; Appendix Fig. S6A–I) showing transient expression of pancreatic endoderm marker *SOX9* and endocrine progenitor marker *NGN3* at stage 4 and stage 5, respectively, increasing expression of *NKX6-1* and *PDX1* from stage 4, and induction of *INS*, *GLP-1R*, *GCG*, *SST*, *NEUROD1*, and *NKX2-2* from stage 5.

Upon ER translocation, the signal peptide is cleaved by signal peptidase from the nascent preproinsulin, generating proinsulin in the ER lumen. The preproinsulin to proinsulin ratio, quantifying the inefficiency of preproinsulin cleavage into proinsulin, was 1.9-fold higher in heterozygous R6C iPSC-derived β cells than in isogenic corrected cells (Fig. 2A,B), but no obvious differences in intermolecular disulfide-linked proinsulin complexes were detected between the two genotypes (Fig. 2C).

To generate more functionally mature iPSC-β cells, we used long-culture differentiation ((Balboa et al, 2022; Virgilio et al, 2025), Fig. EV3A, bottom). Heterozygous R6C and corrected long-cultured iPSC-islets had comparable proinsulin and insulin content (Fig. 2D,E) and proinsulin to insulin ratio, a surrogate marker of impaired proinsulin processing (Fig. 2F). Heterozygous R6C and corrected β cells had comparable static proinsulin secretion (Fig.

3A); 16.8 mM glucose-stimulated insulin secretion was somewhat decreased (stimulation index: 2.3 ± 0.4 vs. 3.9 ± 0.5) and proinsulin to insulin secretion ratio increased in heterozygous R6C cells (Fig. 3B,C). Dynamic insulin secretion in a perifusion setup showed comparable secretion (with a trend for heterozygous R6C β cells to secrete more, Fig. 3D–H). Altogether, heterozygous R6C β cells exhibit impaired preproinsulin translocation and proinsulin conversion, but otherwise roughly preserved insulin biosynthesis and secretion.

## Accumulated preproinsulin in homozygous R6C iPSC-derived β cells leads to reduced proinsulin and insulin content

We then asked whether increasing R6C allele dosage from 50 to 100% would worsen the cellular phenotype. We generated homozygous R6C iPSCs from the proband (Fig. 1; Table 1, A-III-1), corrected the R6C variant using CRISPR/Cas9, and obtained isogenic non-edited iPSCs (Appendix Fig. S7). All iPSC lines passed the aforementioned quality controls (Appendix Figs. S7–S10; Appendix Table S3). Like the heterozygous cells, homozygous R6C and corrected iPSCs differentiated well into similarly viable islets containing β-like cells (Figs. EV2C and EV3D), following a normal developmental pathway (Fig. EV3E; Appendix Fig. S11A–I) with expression of *SOX9* and *NGN3* at stage 4 and stage 5, respectively, increasing expression of *NKX6-1* and *PDX1* from stage 4, and *INS*, *GLP-1R*, *GCG*, *SST*, *NEUROD1*, and *NKX2-2* from stage 5. *NKX6-1* and *PDX1* were higher in homozygous iPSC-islets from stage 4 to 7, and *INS* was also increased (Appendix Fig. S11C,D; Fig. EV3E).

The preproinsulin to proinsulin ratio was 5.3-fold higher in homozygous R6C β cells compared to corrected cells (Fig. 4A,B), and it increased further in the presence of proteasome inhibitor MG132 (Fig. 4C), meaning that up to 2/3 of preproinsulin failed to translocate into the ER. Disulfide-linked proinsulin complexes shifted from smaller disulfide-linked dimers (~15–20 kDa) in corrected cells to higher molecular weight forms (at the top of the gel) in homozygous R6C β cells (Fig. 4D). Long-cultured homozygous R6C iPSC-islets tended to have less proinsulin and had 2.8-fold lower insulin content compared to corrected cells at all stages of maturation, with unchanged proinsulin to insulin ratios (Fig. 4E–H).

## Homozygous R6C iPSC-derived β cells have impaired insulin and proinsulin secretion

In the static secretion setup, homozygous R6C β cells showed around 50% lower proinsulin and insulin secretion compared to corrected cells (Fig. 5A,B), with unchanged proinsulin to insulin ratio (Fig. 5C). In dynamic secretion studies, homozygous R6C β cells had more than twofold lower insulin secretion, basally, at 16.8 mM glucose, 16.8 mM glucose + 50 ng/mL (11.8 nM) exendin-4 and 2.8 mM glucose + 30 mM KCl (Fig. 5D–I). To investigate whether impaired insulin secretion was linked to mitochondrial dysfunction, we assessed mitochondrial respiratory capacity by Seahorse assay. Basal and pyruvate-stimulated respiration was comparable, and maximal respiratory capacity under FCCP challenge was significantly higher in dispersed long-cultured homozygous R6C iPSC-islets (Appendix Fig. S12A–D). In sum,

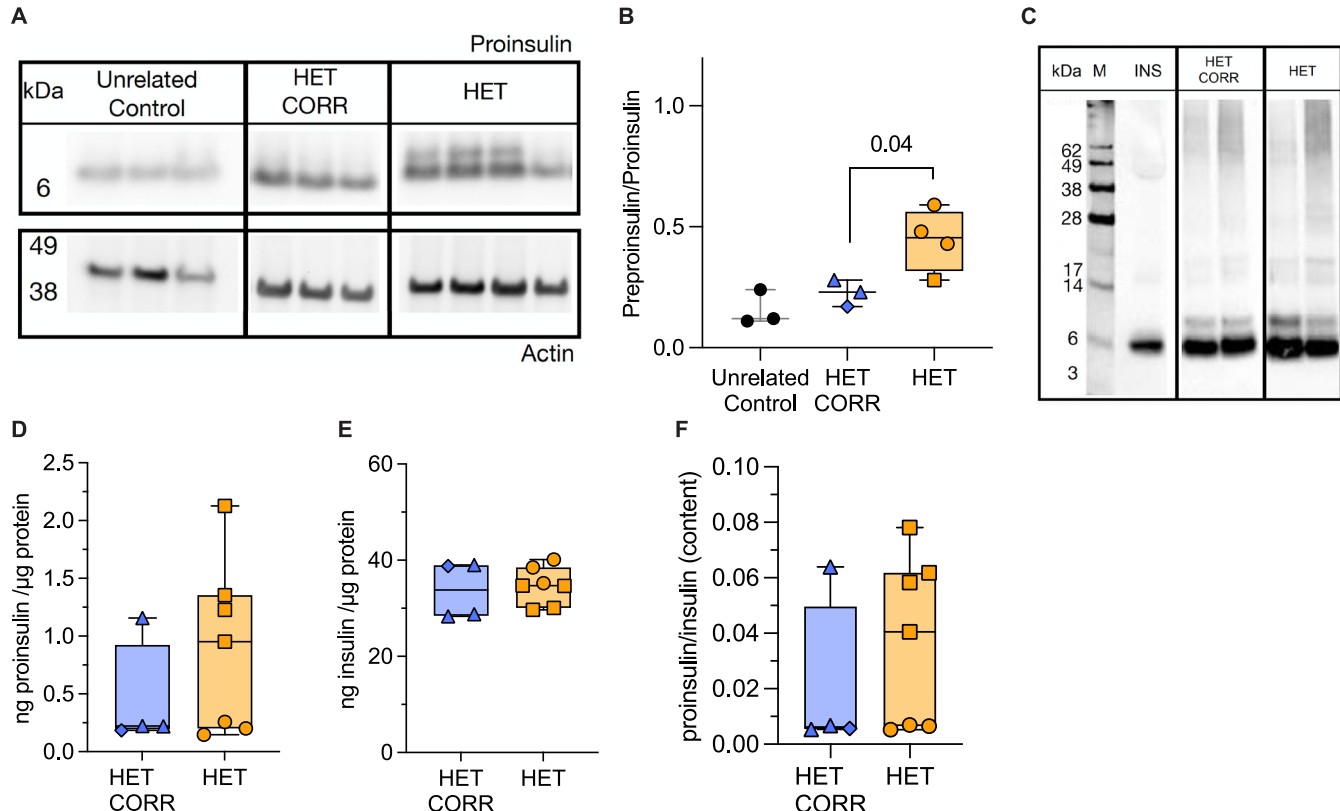

**Figure 2. Heterozygous R6C iPSC-derived β cells have moderate preproinsulin accumulation and normal proinsulin and insulin content.**

Unrelated control (black), heterozygous R6C (HET, yellow) and isogenic corrected (HET CORR, blue) iPSCs were differentiated into β cells. (A) iPSC-islet lysates were analyzed by SDS-PAGE under reducing conditions, electrotransfer to nitrocellulose, and immunoblotting with anti-human proinsulin. The blot was cropped and rearranged for clarity. The left panel is reused in Fig. 4A as it represents a shared control. (B) Quantification of preproinsulin to proinsulin ratio from (A) (unrelated control $n = 3$, HET CORR $n = 3$, HET $n = 4$). (C) iPSC-islet lysates were resolved by nonreducing 12%-NuPAGE, and the completed gel was then treated with 100 mM DTT at 60 °C for 10 min before electrotransfer to nitrocellulose and immunoblotting with anti-proinsulin. $n = 2$. Medium from Min6 β cells transfected with human proinsulin was used as a positive control (INS, lane next to marker, M). The blot was cropped and rearranged for clarity. The left panel is reused in Fig. 4D as it represents the shared marker and positive control. (D) Proinsulin and (E) insulin content (ng) normalized to total protein content (μg). (F) Proinsulin to insulin content ratio from (D, E). (D–F) HET CORR $n = 4$, HET $n = 7$. All panels: Unpaired $t$-test. In box plots, the median of independent experiments is shown by a horizontal line; 25th and 75th percentiles are at the bottom and top of the boxes; whiskers represent the minimum and maximum values. Source data are available online for this figure.

carrying two R6C alleles impairs ER translocation of nascent preproinsulin by two-thirds, reduces basal and stimulated proinsulin and insulin secretion by 50% and increases mitochondrial respiratory capacity.

## Homozygous R6C β cells recapitulate human insulin deficiency in vivo

We next examined in vivo maturation and function of iPSC-islets (Balboa et al, 2022; Virgilio et al, 2025). Homozygous R6C and corrected β cells were purified from stage 7 iPSC-islets by magnetic-activated cell sorting (MACS) using the cell surface marker CD49a (Veres et al, 2019) to 82 ± 3% vs. 84 ± 3% purity. β-cell-purified aggregates were implanted under the kidney capsule of immuno-compromised Rag2 knockout mice (Fig. 6A, (Gorgogietas et al, 2023)). Plasma human C-peptide levels were around or below the detection limit 1 month after transplantation. Fasting and intraperitoneal glucose tolerance test (IpGTT) human C-peptide levels increased gradually over 2, 3, and 4 months of in vivo

maturation in mice transplanted with corrected β cells, but they remained low in mice with homozygous R6C grafts (Fig. 6B,C; Appendix Fig. S13A). Body weight, fasting blood glucose and glucose tolerance remained comparable over 4 months (Fig. 6D; Appendix Fig. S13B–D).

## Insulin deficiency of homozygous R6C β cells is not enhanced by GLP-1 receptor agonists

Functional characterization of the R6C mutation suggests gene dosage-dependent β-cell dysfunction. We then examined the functional activity of insulin molecules secreted by homozygous and heterozygous R6C and isogenic corrected iPSC-islets. HepG2 cells exposed to conditioned medium showed comparable levels of phosphorylated AKT (Protein Kinase B), indicating similar insulin signaling activation (Appendix Fig. S14).

In light of these findings and given the suboptimal glycemic control (HbA1c >8%, Fig. 6E) despite high-dose insulin therapy in the homozygous proband (Fig. 1, A-III-1), we explored adjunctive

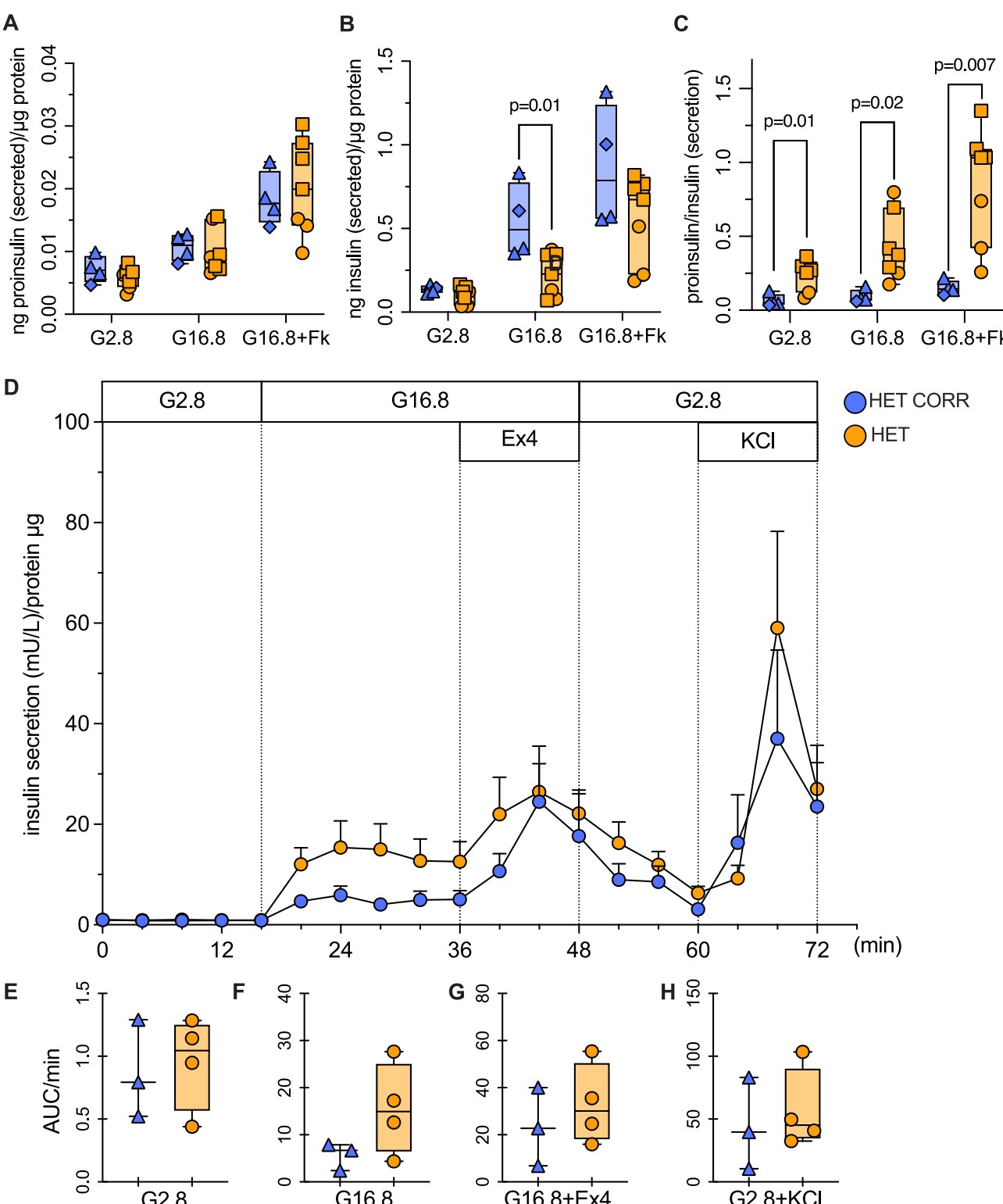

therapeutic strategies. GLP-1 receptor agonists (GLP-1RA) enhance glucose-dependent insulin secretion and suppress inappropriate glucagon release to improve glycemic control (Alfaris et al, 2024). Guided by clinical improvement in the heterozygous father (Fig. 1,

A-II-3) following GLP-1RA therapy (Fig. 6F), we explored the potential of GLP-1RA to preserve homozygous R6C β cells and enhance function. The 48-h treatment of homozygous R6C and corrected stage 7 iPSC-islets with exendin-4, dulaglutide, or

**Figure 3.   Heterozygous R6C iPSC-derived β cells have comparable insulin secretion to corrected cells.**

Heterozygous R6C (HET, yellow) and isogenic corrected (HET CORR, blue) iPSCs were differentiated into long-cultured β cells. (A–C) Static proinsulin and insulin secretion in response to 2.8 mM glucose (G2.8), 16.8 mM glucose (G16.8), or 16.8 mM glucose plus 10 μM forskolin (G16.8 + Fk). HET CORR $n = 4$, HET $n = 7$. (A) Proinsulin and (B) insulin secretion (ng) normalized to protein content (μg). (C) Proinsulin to insulin ratio from (A, B). (D–H) Dynamic insulin secretion upon perifusion with 2.8 mM glucose, 16.8 mM glucose (G16.8), G16.8 plus exendin-4 (Ex4, 50 ng/mL, 11.8 nM) or G2.8 plus KCl (30 mM). HET CORR $n = 3$, HET $n = 4$. (D) Insulin secretion normalized to protein content, with (E–H) area under the curve (AUC) per minute of secretion at G2.8, G16.8, G16.8 + Ex4, and G2.8 + KCl. All panels: Unpaired $t$-test. In box plots, the median of independent experiments is shown by a horizontal line; 25th and 75th percentiles are at the bottom and top of the boxes; whiskers represent the minimum and maximum values. In time course line plots, data are shown as mean ± s.e.m. Source data are available online for this figure.

liraglutide did not alter cell death, rates of which were low (Fig. 6G). GLP-1RA did not induce ER stress markers *BiP (HSPA5)* or *CHOP (DDIT3)*, and, interestingly, tended to reduce the pro-apoptotic proteins *DP5* or *PUMA* (Appendix Fig. S15A–D).

We then evaluated the long-term effect of GLP-1RA on R6C β cell function. After 4 months of in vivo maturation of iPSC-islets (Fig. 6A), mice were twice weekly injected with dulaglutide for 2 months, with treatment washout in the last week. Body weight was not different (Appendix Fig. S13B). Fasting and IpGTT human C-peptide levels increased in corrected mice even after washout, yet they remained negligible in homozygous mice (Fig. 6B; Appendix Fig. S13A,E). Fasting glucose and glucose tolerance improved in both groups after dulaglutide treatment, and this effect was more profound in corrected mice (Fig. 6C; Appendix Fig. S13C,F), in keeping with their improved C-peptide secretion.

These findings indicate that GLP-1RAs exendin-4, dulaglutide, and liraglutide are not β-cell toxic in the context of the homozygous *INS* R6C mutation. While GLP-1RAs may serve as a supportive intervention, they are unlikely to restore β-cell function in the presence of homozygous R6C mutation and emphasize the need for genotype-tailored therapeutic strategies.

## Repression of protein translation and translocation processes in homozygous R6C β cells

Considering the potential role for ER stress in insulin signal peptide mutations (Liu et al, 2012; Meur et al, 2010), we further investigated the molecular underpinnings of impaired insulin production in R6C β cells. GFP-sorted wildtype or homozygous R6C insulin plasmid-transfected *INS*-knockout EndoC-βH1 cells had similar *BiP* and *CHOP* expression (Appendix Fig. S15E,F). We stressed homozygous, heterozygous, and isogenic control stage 7 iPSC-islets with synthetic ER stressors tunicamycin (an inhibitor of N-linked glycosylation, 48 h), thapsigargin (an inhibitor of the sarco/endoplasmic reticulum $Ca^{2+}$ ATPase, 48 h) and brefeldin A (an inhibitor of ER-Golgi transport, 24 h). All stressors induced cell death, without differences between R6C and isogenic control islets (Fig. EV2B,C). Thapsigargin and brefeldin A induced *BiP*, *CHOP*, *DP5* and *PUMA* in homozygous and corrected iPSC-islets (Appendix Fig. S16A–H). GLP-1RAs protected homozygous R6C iPSC-islets from the ER stressors (Appendix Fig. S17A,B) without obvious changes in *BiP*, *CHOP*, *DP5* and *PUMA* (Appendix Fig. S16A–H). These results refute ER stress as the driving force of R6C pathogenesis yet indicate that GLP-1RAs confer protection.

To broadly examine gene expression, we performed bulk-RNA sequencing of MACS-purified stage 7 β cells (Fig. 7A). Of the 35 differentially expressed genes (FDR <0.05, |$\log_2$ fold change| >0.58) (Appendix Table S5), the majority displayed low expression levels

(<1 TPM), possibly reflecting transcriptional noise at low counts. We manually curated seven small, focused gene sets—β cell identity, ER translocation, ER-associated degradation & pre-emptive quality control, unfolded protein response, ER-Golgi transport, insulin processing, insulin secretion, and cytosolic stress (Appendix Table S6). Gene-wise Z-scores (Wang et al, 2024) for each category were not different between homozygous R6C and corrected β cells (Fig. 7B). Gene set enrichment analysis (Korotkevich et al, 2021; Subramanian et al, 2005; Fabregat et al, 2018; Kanehisa and Goto, 2000; Ashburner et al, 2000; Liberzon et al, 2015) revealed an overall repressed profile in homozygous R6C cells (Fig. 7C; Dataset EV1), including downregulation of translation initiation and elongation, SRP dependent co-translational protein targeting to membrane, integrated stress response, ERK1 and ERK2 cascade, glycolysis, apoptosis signaling, and immune responses (e.g., interferon signaling and response, interleukin signaling, etc.). Upregulated pathways included pancreatic β-cell signatures, tRNA synthesis, and butyrate metabolism. *INS* expression was stable (Figs. 7D and EV3E). In conclusion, homozygous R6C β cells have a disrupted transcriptional landscape, characterized by global repression of translation, immune and inflammatory responses, and ER translocation, alongside subtle shifts in pathways critical for insulin processing and secretion.

## Discussion

This study provides a unified mechanistic and clinical framework for understanding the R6C signal peptide variant in the insulin gene. We identified *INS* R6C homozygosity as the cause of a new recessive form of monogenic diabetes. By integrating clinical observations, patient-derived iPSC-β cell models, transcriptomic profiling, and population-level data, our results suggest reduced penetrance of the R6C variant. Disrupted signal peptide integrity leads to severely reduced insulin biosynthesis and secretion in homozygous R6C β cells, which translates into early-onset, insulin-requiring diabetes. In contrast, one wild-type insulin allele in heterozygous R6C carriers compensates for this defect, consistent with variable adult-onset hyperglycemia or normoglycemia. These findings revise the original classification of R6C as a dominant variant and provide a blueprint for functionally reassessing other variants of uncertain significance in monogenic diabetes.

The increasing availability of large-scale population datasets is reshaping how we interpret the pathogenicity of rare variants. R6C shows a frequency incompatible with penetrant monogenic diabetes. The absence of homozygous carriers in large datasets and early-onset, insulin-dependent diabetes in the presently reported families validates a recessive model.

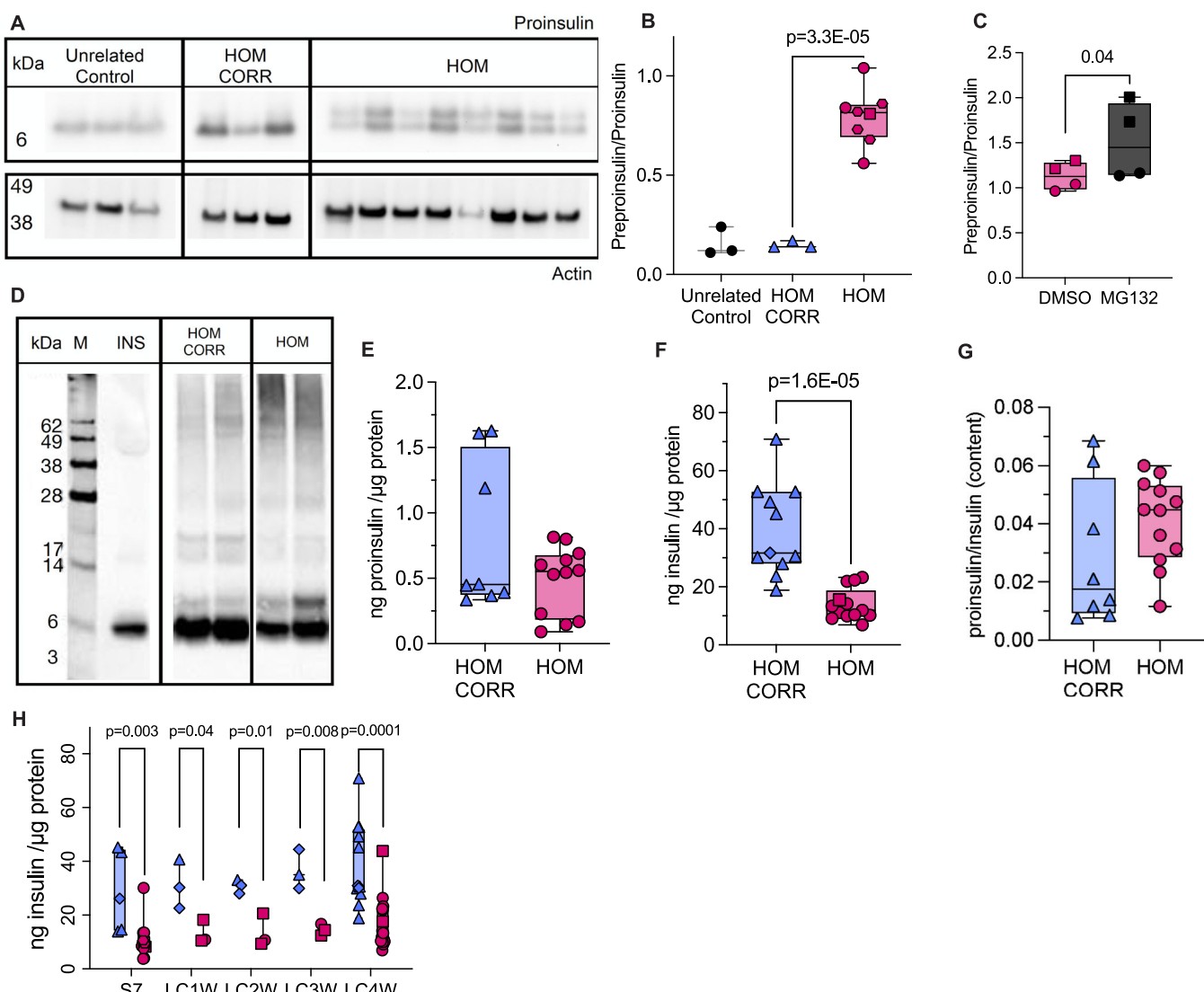

**Figure 4. Accumulated preproinsulin in homozygous R6C iPSC-derived β cells leads to reduced proinsulin and insulin content.**

Unrelated control (black), homozygous R6C (HOM, pink) and isogenic corrected (HOM CORR, blue) iPSCs were differentiated into β cells. (A) iPSC-islet lysates were analyzed by SDS-PAGE under reducing conditions, electrotransferred to nitrocellulose, and immunoblotted with anti-human proinsulin. The blot was cropped and rearranged for clarity. The left panel is reused as in Fig. 2A, as it represents a shared control. (B) Quantification of preproinsulin to proinsulin ratio from (A) (unrelated control $n = 3$, HOM CORR $n = 3$, HOM $n = 8$) and (C) of homozygous R6C iPSC-β cells treated with vehicle DMSO or MG132 (10 μM) for 30 min ($n = 4$). (D) iPSC-islet lysates were resolved by nonreducing 12%-NuPAGE and the completed gel then treated with 100 mM DTT at 60 °C for 10 min before electrotransfer to nitrocellulose and immunoblotting with anti-proinsulin. $n = 2$. Medium from Min6 β cells transfected with human proinsulin was used as a positive control (INS, lane next to marker, M). The blot was cropped and rearranged for clarity. The left panel is reused as in Fig. 2C, as it represents a shared marker and a positive control. (E) Proinsulin and (F) insulin content normalized to total protein content. (G) Proinsulin to insulin content ratio from (E, F). (E–G) HOM CORR $n = 8$, HOM $n = 12$–13. (H) Insulin content normalized to total protein content along stage 7 (S7, HOM CORR $n = 5$, HOM $n = 12$), 1 week (LC1W, HOM CORR $n = 3$, HOM $n = 3$), 2 weeks (LC2W, HOM CORR $n = 3$, HOM $n = 3$), 3 weeks (LC3W, HOM CORR $n = 3$, HOM $n = 3$), and 4 weeks (LC4W) of long culture (LC, HOM CORR $n = 12$, HOM $n = 15$). All panels: Unpaired $t$-test. In box plots, the median of independent experiments is shown by a horizontal line; 25th and 75th percentiles are at the bottom and top of the boxes; whiskers represent the minimum and maximum values. Source data are available online for this figure.

Patients' iPSC-β cells provided the cellular and molecular lens through which we observed the R6C impact: disrupted preproinsulin ER translocation in homozygous R6C β cells, preproinsulin accumulation and insulin storage and secretion collapse. These findings agree with the in silico model showing impaired SRP54 interaction (Gutierrez Guarnizo et al, 2023) and inverted signal peptide insertion in the SEC61 translocon, reducing ER translocation efficiency. Interestingly, a recent study showed increased photo-crosslinking intensities between the R6C signal peptide and SRP due to increased h-region hydrophobicity (Miller et al, 2024), which could potentially explain less efficient SRP54 to SEC61 handover. However, this disagrees with our in silico prediction of slightly impaired hydrogen bond and protein-protein interaction, underlining that machine learning tools and structural prediction algorithms need to be paired with cellular phenotyping to refine classification frameworks for rare variants.

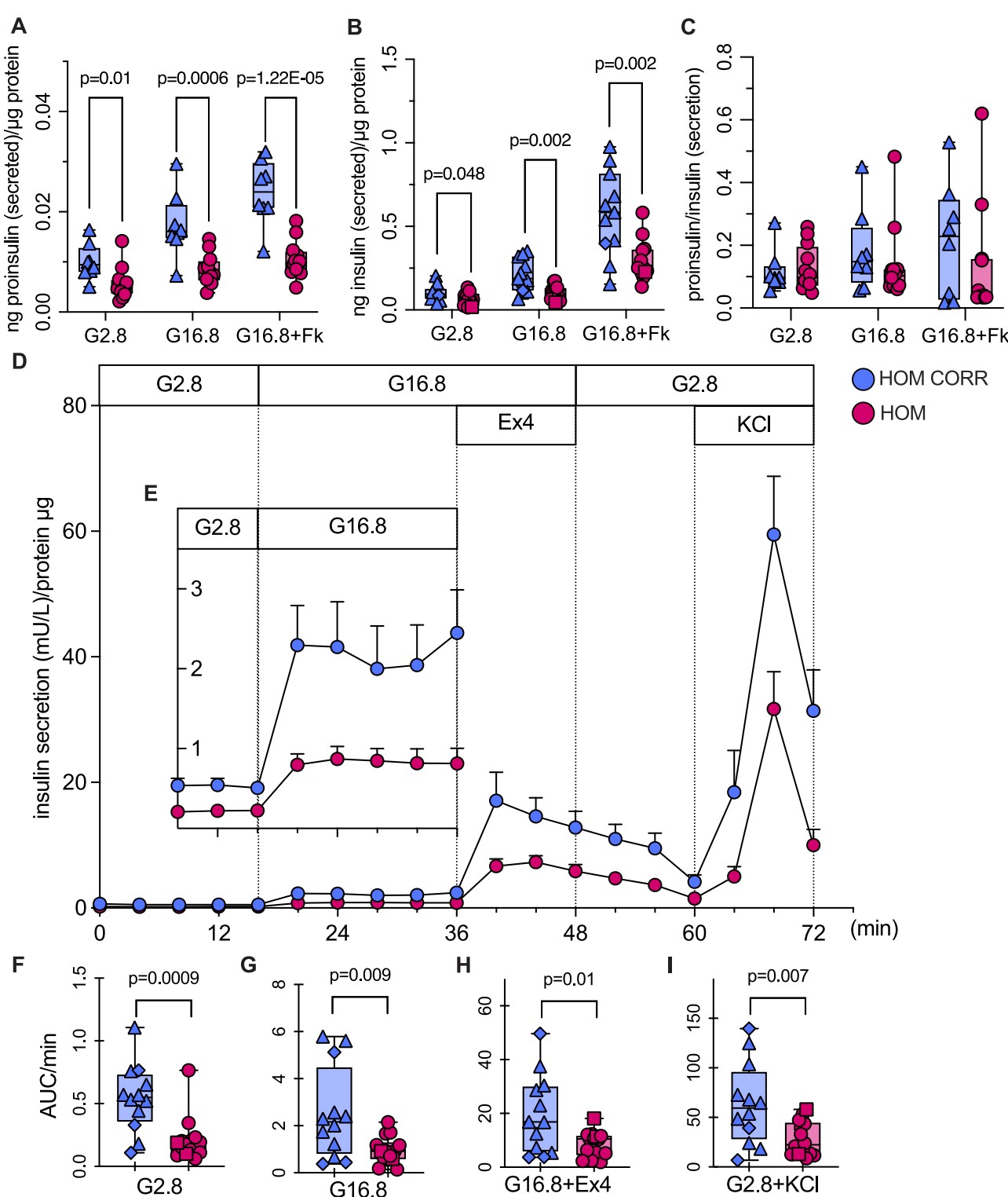

Downregulated translation initiation/elongation in homozygous R6C β cell transcriptomes indicates that these cells potentially engaged an adaptive proteostasis program to attenuate global protein synthesis and limit further accumulation of preproinsulin.

This attenuation is at the level of translation, as insulin mRNA was not reduced. A recent study showed that β cells favor translation and translocation of preproinsulin bearing a strong signal peptide (wild-type R6) against a weak one (R6H), without insulin

◄ **Figure 5.  Homozygous R6C iPSC-derived β cells have impaired proinsulin and insulin secretion.**

Homozygous R6C (HOM, pink) and isogenic corrected (HOM CORR, blue) iPSCs were differentiated into long-cultured β cells. (A–C) Static proinsulin and insulin secretion in response to 2.8 mM glucose (G2.8), 16.8 mM glucose (G16.8), or 16.8 mM glucose plus 10 μM forskolin (G16.8 + Fk). HOM CORR $n = 8$, HOM $n = 12$ (for proinsulin), HOM $n = 13$ (for insulin). (A) Proinsulin and (B) insulin secretion normalized to protein content. (C) Proinsulin to insulin ratio from (A, B). (D–I) Dynamic insulin secretion upon perifusion with 2.8 mM glucose, 16.8 mM glucose (G16.8), G16.8 plus exendin-4 (Ex4, 50 ng/mL, 11.8 nM), or G2.8 plus KCl (30 mM). HOM CORR $n = 12$, HOM $n = 14$. (D) Insulin secretion normalized to protein content, with (E) zoom in on 16.8 mM glucose response. (F–I) Area under the curve (AUC) per minute of secretion at G2.8, G16.8, G16.8 + Ex4 and G2.8 + KCl. All panels: Unpaired $t$-test. In box plots, the median of independent experiments is shown by a horizontal line; 25th and 75th percentiles are at the bottom and top of the boxes; whiskers represent the minimum and maximum values. In time course line plots, data are shown as mean ± s.e.m. Source data are available online for this figure.

mRNA differences (Xu et al, 2024). At the mRNA level, there was also downregulation of SRP binding and targeting in homozygous cells, due to less incoming nascent preproinsulin to the ER. We did not observe elevated ER stress in homozygous cells, in keeping with non-human or non-β-cell models (Xu et al, 2024; Guo et al, 2014). We propose that the lack of efficient ER translocation of nascent preproinsulin prohibits their interaction with BiP and translocon-associated proteins (Xu et al, 2022; Li et al, 2019; Haßdenteufel et al, 2018), thus avoiding ER stress. This creates a proteostatic "dormant" niche that preserves viability at the expense of insulin-secretory function (Rutkowski et al, 2006). Cellular adaptation to persistent stress is further suggested by the increased maximal respiration capacity, mirroring findings in cytosolic α-synuclein-aggregating cells in Parkinson's disease (Schirinzi et al, 2022; Annesley et al, 2016) and islet amyloid polypeptide-rich INS-1E cells (Soty et al, 2011). Doxycycline-inducible R6C overexpression increased cell death and cytosolic stress with upregulated HSP70 (Guo et al, 2014); this was not observed in our *INS*-knockout EndoC-βH1 complementation model. Adult mice with conditional tamoxifen-induced deletion of *Ins2* on an *Ins1*-null background showed alleviation of ER stress and β-cell proliferation prior to overt hyperglycemia (Szabat et al, 2016). These effects are more likely the result of acute interruptions in cellular processes or an overload of proteotoxic stress. The patient-derived iPSC-islet model may reflect a more physiologically relevant, gradually acquired homeostatic state in which adaptive mechanisms limit acute cytotoxicity. In the long term, however, little C-peptide is detected in the in vivo R6C β cell model and homozygous R6C carriers do develop diabetes, demonstrating that these compensatory responses are insufficient to sustain insulin secretion under chronic metabolic demand, and that progressive insulin deficiency drives the homozygous R6C diabetes phenotype.

Heterozygous cells, by contrast, maintain sufficient SRP binding and translocation, showing no sign of wild-type proinsulin trapping by R6C mutant proinsulin and largely normal insulin secretion. These findings align with population-level data that heterozygous carriers do not show increased risk of diabetes. Long-term follow-up is essential to assess disease progression in diabetic carriers as well as lifetime diabetes risk among unaffected individuals. The reduced penetrance of R6C and other forms of monogenic diabetes (Li et al, 2023a; Mirshahi et al, 2022; Goodrich et al, 2021) suggests the influence of environmental and/or genetic modifiers (Kim et al, 2003; Kristinsson et al, 2001; Alam et al, 2021). Metabolic and environmental stressors such as obesity, sedentary lifestyle, or

pregnancy could increase β-cell workload, unmasking otherwise compensated insulin biosynthesis defects in heterozygotes at a later age (Bonnefond and Semple, 2022; Mittendorfer et al, 2023; James and Stanford, 2025). In the current study, the heterozygous father (A-II-3) developed diabetes with a BMI of 39 kg/m², and the heterozygous mothers (A-II-4, B-I-2) had gestational diabetes, glucose intolerance or diabetes treated with oral glucose-lowering agents, suggesting metabolic demand accelerated diabetes onset. This metabolic demand hypothesis fits with another insulin p.Leu30Met variant (L30M, *INS* c.88 C > G) resulting in early-onset diabetes. Notably, a heterozygous L30M carrier (BMI 21.7 kg/m²) maintained normoglycemia until age 68 years (Meur et al, 2010). Heterozygote individuals may remain normoglycemic until stress-related pressure reveals the defect, and/or genetic background may modulate disease penetrance (George et al, 2021; Kingdom et al, 2024). Polygenic burden, arising from common variants, can indeed modulate the penetrance and severity of monogenic diabetes (Luckett et al, 2025; Huerta-Chagoya et al, 2024; Leech et al, 2025). Future whole-genome sequencing studies and genetic risk score evaluations in large families or cohorts with variable heterozygous phenotypes may help identify protective or sensitizing alleles.

Monogenic diabetes requires tailored therapeutic strategies. Good glycemic control was not achieved in three out of four homozygous R6C patients. Insulin secreted by homozygous, heterozygous and corrected control iPSC-islets showed similar activation of insulin signaling in liver cells, suggesting normal functionality of R6C insulin. Our in vivo experiments demonstrate that GLP-1RA transiently improved glycemia, not by correcting the insulin secretory defect but possibly via glucagon suppression, delayed gastric emptying, or improved insulin sensitivity (Drucker, 2018). These results suggest that homozygous R6C patients may require a combination of therapies to achieve good control.

By integrating population genetics and patient-derived iPSC models, we redefine *INS* R6C as a recessive mutation. Our work not only resolves a persistent clinical ambiguity but also establishes a generalizable framework for functionally classifying rare diabetes alleles. Going forward, identifying genetic and/or environmental modifiers of R6C penetrance and evaluating therapies that restore signal-peptide translocation will be vital for translating these insights into patient care. Moreover, applying proteomics, metabolomics, and spatial transcriptomics will illuminate how signal-peptide defects converge with broader β-cell stress pathways and uncover shared targets for optimized intervention.

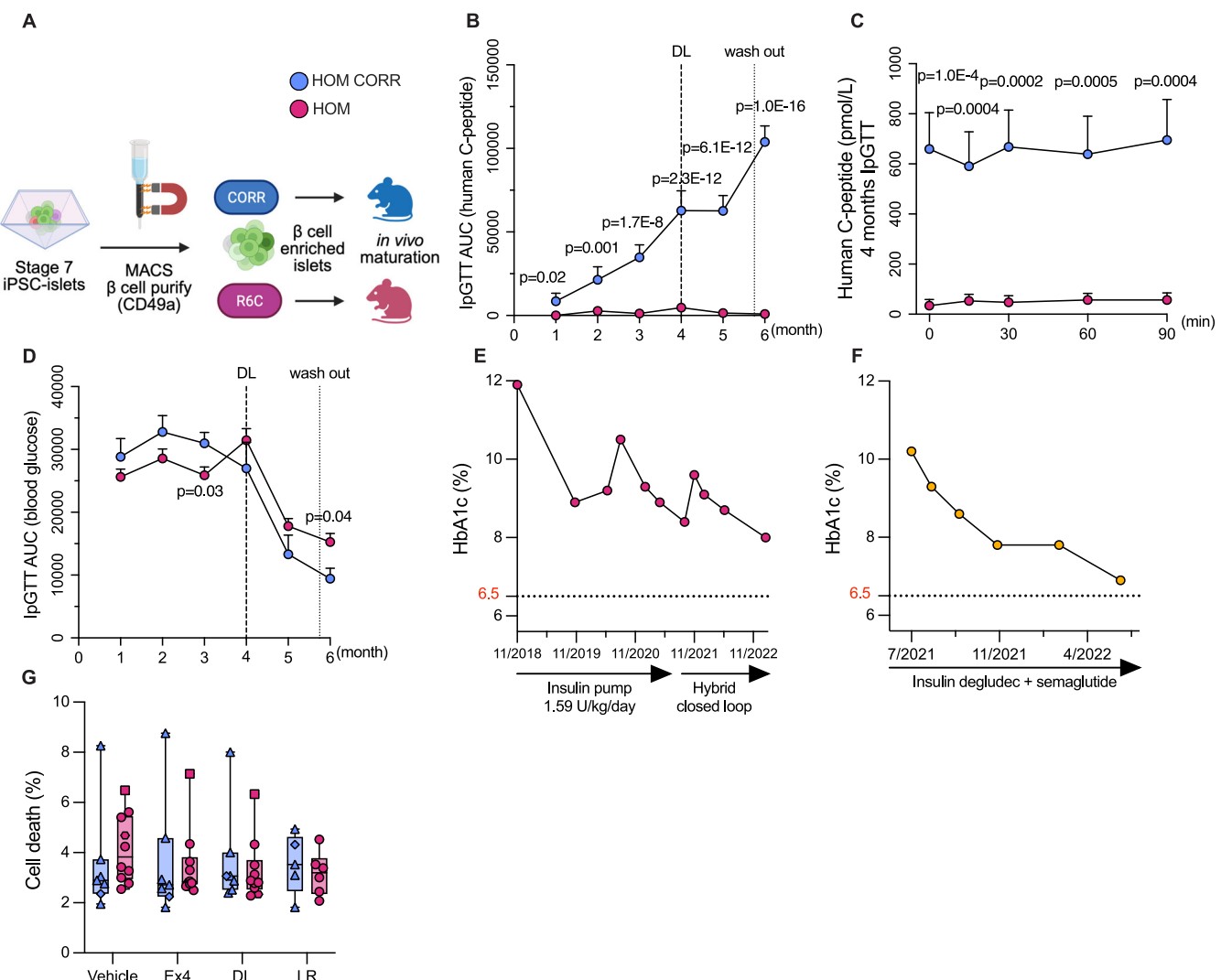

**Figure 6. Humanized homozygous R6C mice secrete minimal human C-peptide.**

Homozygous R6C (HOM, pink) and isogenic corrected (HOM CORR, blue) MACS-purified iPSC-β cell aggregates were transplanted under the kidney capsule of immunocompromised Rag2 knockout mice and followed up for 4 months prior to a 2-month treatment with dulaglutide (DL, 1 mg/kg, twice weekly, dashed line). The last week (dotted line) represents the washout of treatment. (A) Scheme of in vivo experiment. (B) Area under the curve (AUC) of plasma human C-peptide levels during an intraperitoneal glucose tolerance test (IpGTT) from 1 month after transplantation to 2-month dulaglutide treatment. Mixed-effects analysis with Tukey correction for multiple comparison. (C) Human C-peptide levels during an IpGTT at 4 months after transplantation. Unpaired t-test with Holm–Šidák correction for multiple comparison (HOM CORR $n = 6$, HOM $n = 6$). (D) AUC of blood glucose levels during IpGTTs from 1 month after transplantation to 2-month dulaglutide treatment. Mixed-effects analysis with Tukey correction for multiple comparison. For (B, D), sample sizes (n, HOM CORR vs. HOM) for each timepoint were: 1 month (7 vs. 17), 2 months (9 vs. 19), 3 months (7 vs. 17), 4 months (7 vs. 9), 5 months (3 vs. 8), and 6 months (3 vs. 7). HbA1c and treatment of (E) homozygous R6C proband (Fig. 1, A-III-1) and (F) heterozygous father (Fig. 1, A-II-3). (G) Cell death (%) of stage 7 iPSC-islets treated in vitro for 48 h with glucagon-like peptide-1 receptor agonists exendin-4 (Ex4, 50 nM), dulaglutide (DL, 50 nM), and liraglutide (LR, 50 nM) or vehicle PBS (HOM CORR $n = 10$, HOM $n = 7$), individual conditions contained missing observations. In box plots, the median of independent experiments is shown by a horizontal line; 25th and 75th percentiles are at the bottom and top of the boxes; whiskers represent the minimum and maximum values. In time course line plots, data are shown as mean ± s.e.m. Source data are available online for this figure.

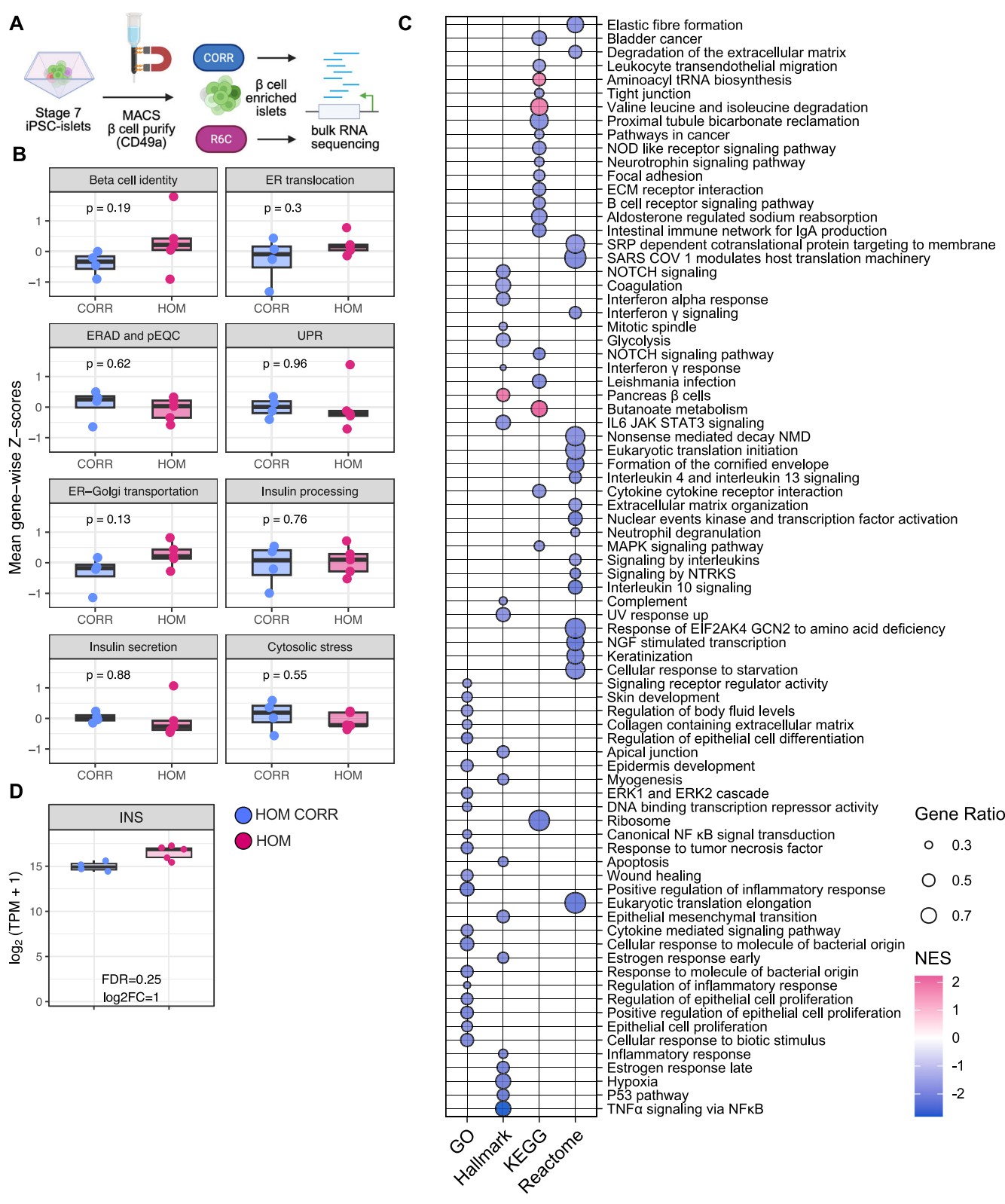

◀ **Figure 7. Homozygous R6C β cell transcriptomes demonstrate repression of protein translation and ER-related processes.**

(A) Scheme of bulk-RNA sequencing of stage 7 MACS-purified iPSC-β cells. (B) Gene-wise Z-score comparing relative gene expression levels between homozygous R6C and isogenic corrected β cells. Unpaired t-test. (C) Bubble plot of enriched gene sets or pathways in GSEA analysis on GO, Hallmark, KEGG and Reactome (FDR <0.05, top 20 of each set). Gene ratio (dot size): represents the proportion of input genes associated with a given pathway relative to the total number of input genes mapped. Normalized enrichment score (NES, dot color): accounts for gene set size and adjusts the enrichment score to allow cross-comparison between gene sets. Positive scores (pink) indicate enrichment in the upregulated set, and negative scores (blue) indicate enrichment in the downregulated set. (D) $\log_2(TPM + 1)$ of *INS* gene (TPM: transcripts per kilobase million). Differential expression analysis was performed using DESeq2, which models count data with negative binomial dispersion estimates. $P$ values were adjusted for multiple testing using the Benjamini–Hochberg false discovery rate (FDR). $n = 3$ independent differentiations for each group. In box plots, the median of independent experiments is shown by a horizontal line; $25^{th}$ and $75^{th}$ percentiles (interquartile range, IQR) are at the bottom and top of the boxes; whiskers represent the most extreme values within 1.5 × IQR from the hinges. Data points outside this range are displayed individually as outliers.

# Methods

### Reagents and tools table

| Reagent/resource | Reference or source | Identifier or catalog number |
| --- | --- | --- |
| **Experimental models** | | |
| EndoC-βH1 (*H. sapiens*) | Raphaël Scharfmann Lab (Ravassard et al, 2011) | NA |
| Peripheral blood mononuclear cells of individual A-III-1 (Table 1) (*H. sapiens*) | Pediatric Endocrinology and Diabetology (DECCP), Centre Hospitalier de Luxembourg | NA |
| Peripheral blood mononuclear cells of individual A-II-3 (Table 1) (*H. sapiens*) | Pediatric Endocrinology and Diabetology (DECCP), Centre Hospitalier de Luxembourg | NA |
| *INS* c.16 C > T heterozygous induced pluripotent stem cell (iPSC) line ULBi011.5 (*H. sapiens*) | This study | NA |
| *INS* c.16 C > T heterozygous iPSC line ULBi011.5-CR29 (non-edited CRISPR control) (*H. sapiens*) | This study | NA |
| *INS* c.16 C > T heterozygous corrected iPSC line ULBi011.5-CR21 (*H. sapiens*) | This study | NA |
| *INS* c.16 C > T heterozygous corrected iPSC line ULBi011.5-CR30 (*H. sapiens*) | This study | NA |
| *INS* c.16 C > T homozygous iPSC line ULBi012.8 (*H. sapiens*) | This study | NA |
| *INS* c.16 C > T homozygous iPSC line ULBi012.31 (*H. sapiens*) | This study | NA |
| *INS* c.16 C > T homozygous iPSC line ULBi012.31-CR4 (non-edited CRISPR control) (*H. sapiens*) | This study | NA |
| *INS* c.16 C > T homozygous corrected iPSC line ULBi012.31-CR3 (*H. sapiens*) | This study | NA |

| Reagent/resource | Reference or source | Identifier or catalog number |
| --- | --- | --- |
| *INS* c.16 C > T homozygous corrected iPSC line ULBi012.31-CR41 (*H. sapiens*) | This study | NA |
| Rag2 knockout mice (*M. musculus*) | The Jackson Laboratory | B6.Cg-Rag2tm1.1Cgn/J |
| HepG2 (*H. sapiens*) | Raphaël Scharfmann Lab | NA |
| MIN-6 (*M. musculus*) | Dr. D. Stoffers | NA |
| **Recombinant DNA** | | |
| Insulin pRIG plasmids | Raphaël Scharfmann Lab | NA |
| SeVdp(KOSM)302 L | (Nishimura et al, 2017) | 170523-1 |
| **Antibodies** | | |
| Rabbit anti-human OCT4 | Cell Signaling Technology | Cat# 2840, RRID:AB_2167691 |
| Mouse anti-human TRA1-60 | Thermo Fisher Scientific | Cat#MA1-023, RRID: AB_2536699 |
| Mouse anti-human SSEA4 | Thermo Fisher Scientific | Cat#MA1-021, RRID: AB_2536687 |
| Rabbit anti-human Nanog | Cell Signaling Technology | Cat#4903, RRID:AB_10559205 |
| Goat anti-human SOX17 | R&D Systems | Cat# AF1924, RRID:AB_355060 |
| Rabbit anti-human vimentin | Abcam | Cat#137321, RRID:NA |
| Mouse anti-human beta tubulin III | Promega | Cat#G7121, RRID:AB_430874 |
| Mouse anti-human NKX6.1 | BD Biosciences | Cat# 563022, RRID:AB_2737958 |
| Goat anti-human PDX1 | R&D Systems | Cat# AF2419, RRID:AB_355257 |
| Guinea pig anti-human INS | Dako | Cat# A0564; RRID:AB_10013624 |
| Mouse anti-human GCG | Sigma-Aldrich | Cat# G2654; RRID:AB_259852 |
| Rabbit anti-human SST | Abcam | Cat#ab108456; RRID: AB_11158517 |
| Mouse anti-human proinsulin B-C junction sequence KTRREAEDLQ | Abmart | Cat# B-C junction; RRID: AB_2921300 |
| Anti-CD49a antibody PE conjugated | BD Bioscience | Cat#559596; RIDD: AB_397288 |
| Anti-PE microbeads | Miltenyi | Cat#130-105-639; RIDD:NA |

| Reagent/resource | Reference or source | Identifier or catalog number |
|---|---|---|
| Mouse anti-human β-Actin (HepG2) | Proteintech | Cat# CL555-66009; RRID:AB_2919667 |
| Rabbit anti-human p-AKT (HepG2) | Cell Signaling Technology | Cat#4060S; RRID:AB_2315049 |
| Rabbit anti-human AKT (HepG2) | Cell Signaling Technology | Cat#4685S; RRID:AB_2225340 |
| anti-human β-Actin (HepG2) | Sigma | Cat#A5441, RRID:AB_476744 |
| **Oligonucleotides and other sequence-based reagents** | | |
| PCR primers | This study | Appendix Table S8 |
| R6C correction guide RNA | This study | Appendix Table S3 |
| R6C correction homology-directed repair template | This study | Appendix Table S3 |
| **Chemicals, Enzymes and other reagents** | | |
| Alt-R CRISPR Cas9 tracrRNA ATTO 550 | Integrated DNA Technologies | Cat#1075928 |
| Cas9 Electroporation enhancer | Integrated DNA Technologies | Cat# 1075916 |
| Alt-R HDR Enhancer V2 | Integrated DNA Technologies | Cat# 10007921 |
| Alt-R S.p. HiFi Cas9 Nuclease V3 | Integrated DNA Technologies | Cat# 1081061 |
| DMSO | Sigma-Aldrich | Cat# D2650 |
| Exendin-4 | Sigma-Aldrich | Cat# E7144 |
| Dulaglutide | Eli Lilly | Trulicity 1.5 mg |
| Liraglutide | Novo Nordisk | Victoza 1.2 mg |
| Tunicamycin | Sigma-Aldrich | Cat# T7765 |
| Thapsigargin | Sigma-Aldrich | Cat# T9033 |
| Brefeldin A | Sigma-Aldrich | Cat# B6542 |
| Propidium iodide | Sigma-Aldrich | Cat# P4170 |
| Hoechst 33342 | Sigma-Aldrich | Cat# B2261 |
| iProof High-Fidelity DNA Polymerase | Bio-Rad | Cat# 1725301 |
| Forskolin | Sigma-Aldrich | Cat# F6886 |
| NovoRapid | Novo Nordisk | NA |
| **Software** | | |
| GraphPad Prism version 10 | GraphPad Software | NA |
| RStudio version 2024.12.1 | Posit | NA |
| R version 4.4.3 | The R Foundation for Statistical Computing | NA |
| FastQC version 0.12.1 | (Andrews, 2023) | NA |
| fastp version 0.23.2 | (Chen et al, 2018) | NA |
| Salmon version 1.10.3 | (Patro et al, 2017) | NA |
| tximeta version 1.22.1 | (Love et al, 2020) | NA |
| DESeq2 version 1.44.0 | (Love et al, 2014) | NA |
| fgsea version 1.33.1 | (Korotkevich et al, 2021) | NA |
| G*Power version 3.1 | (Faul et al, 2009) | NA |
| FlowJo version 10.7 | https://www.flowjo.com/ | NA |

| Reagent/resource | Reference or source | Identifier or catalog number |
|---|---|---|
| Fiji version 1.54p | https://imagej.net/software/fiji/ | NA |
| SavvyHomozygosity | https://github.com/rdemolgen/SavvySuite | NA |
| UCSF ChimeraX version 1.9 | https://www.cgl.ucsf.edu/chimerax/ | NA |
| AlphaFold 3 | https://www.alphafoldserver.com | NA |
| CellProfiler version 4 | https://www.cellprofiler.org/ | NA |
| Benchling | https://www.benchling.com/ | NA |
| CRISPOR | http://crispor.tefor.net/ | NA |
| BioRender | https://www.biorender.com | NA |
| **Other** | | |
| QIAamp DNA Micro Kit | Qiagen | Cat# 51306 |
| Bio-Rad ChemiDoc | Bio-Rad | NA |
| PERI5LM perifusion system | BioRep, Miami | PERI5LM |
| Human Insulin ELISA Kit | Mercodia | Cat# 10-1113-10 |
| Human Proinsulin ELISA Kit | Mercodia | Cat# 10-1118-01 |
| Human Ultrasensitive C-peptide ELISA | Mercodia | Cat# 10-1141-01 |
| XFp Extracellular Flux Analyzer | Agilent | NA |
| iBLOT Dry Blotting System | Thermo Fisher Scientific | NA |
| iBright imager | Invitrogen | NA |
| Dynabeads mRNA DIRECT Purification Kit | Invitrogen | Cat# 61012 |
| Reverse Transcriptase Core Kit | Eurogentec | Cat# RT-RTCK-03 |
| SsoAdvanced Universal SYBR Green Supermix | Bio-Rad | Cat# 1725274 |
| CFX Connect | Bio-Rad | |
| RNeasy Plus Micro Kit | Qiagen | Cat# 74034 |
| Bioanalyzer RNA 6000 Nano assay | Agilent | Cat# 5067-1511 |
| Agilent 2100 Bioanalyzer | Agilent | |
| NEBNext Ultra II Directional RNA Library Prep Kit for Illumina | New England BioLabs | Cat# E7760L |
| Novogene NGS Stranded RNA Library Prep Set | Novogene | Cat# PT044 |
| Illumina NovaSeq 6000 | Illumina | NA |
| Illumina NovaSeq X plus | Illumina | NA |
| RNeasy Micro Kit | Qiagen | Cat# 74004 |
| First-Strand cDNA Kit | Thermo Fisher Scientific | Cat# K1612 |
| Power SYBR Green mix | Invitrogen | Cat# 4368708 |
| QuantStudio 3 analyzer | Thermo Fisher Scientific | NA |
| Axio Imager A1 | Carl Zeiss | 3521000414 |

| Reagent/resource | Reference or source | Identifier or catalog number |
|---|---|---|
| Axio Observer 7 | Carl Zeiss | 4919170001000 |
| Neon Transfection System | Thermo Fisher Scientific | NA |

## Methods and protocols

### Homozygosity mapping

We used SavvyHomozygosity (Wakeling, 2018; Laver et al, 2022) to perform homozygosity assessment using off-target reads from a targeted next-generation sequencing panel in the proband, and confirmed homozygous-only variants in the *INS* gene targeted region.

### Diabetes association in the UK Biobank and the Geisinger cohort

We used data from the UK Biobank, an ethically approved population cohort of over 500,000 individuals from the UK with whole-genome sequencing and deep phenotypic information (Li et al, 2023b) to assess the diabetes association with *INS* R6C variant carriers. We also used a US health-system-based cohort of 173,247 individuals from Pennsylvania with whole-exome sequencing and deep phenotyping from electronic health records. In both cohorts, diabetes was defined as having one or more of the following criteria: self-reported, having an ICD9/10 code for diabetes, being on a diabetes treatment, or having HbA1c ≥48 mM/M before recruitment. We performed Fisher's exact test to assess the association of diabetes with *INS* R6C and R6H variants.

### In silico modeling of protein structure and interaction

Protein structure modeling was performed using AlphaFold 3 to predict the complex structure (Abramson et al, 2024). The best models were analyzed in ChimeraX v1.9 (Pettersen et al, 2021) for visualization, hydrogen bond detection, and calculation of buried surface area at the binding interface.

### EndoC-βH1 cells

The human pancreatic β cell line EndoC-βH1 (Ravassard et al, 2011) was cultured on plates coated with 1.2% Matrigel and 3 μg/ml fibronectin (both from Sigma-Aldrich) at 37 °C in 5% $CO_2$ in Advanced DMEM/F-12 medium (Thermo Fisher Scientific) supplemented with 2% bovine serum albumin fraction V (Roche), 6.7 ng/ml sodium selenite, 10 mM nicotinamide (Calbiochem), 50 μM β-mercaptoethanol (Sigma-Aldrich), 100 U/ml penicillin, and 100 μg/ml streptomycin (Thermo Fisher Scientific). INS-knockout cells were generated via CRISPR/Cas9 technology (IDT); these cells are to be described in an upcoming article.

### Plasmid cloning, transfection, and FACS

Wildtype and R6C mutated *INS* cDNA (full length) was synthesized (Eurofins genomics) and cloned downstream of a CMV promoter and upstream of an IRES-GFP cassette in an in-house engineered pRIG plasmid (Albagli-Curiel et al, 2007). Wildtype, R6C or both *INS* pRIG plasmids were transfected into *INS*-knockout EndoC-βH1 cells using Lipofectamine 3000 (Thermo Fisher Scientific) according to the manufacturer's instructions. Empty plasmids without *INS* cDNA but with IRES-GFP were used as control. At day 1 and 3 post transfection, cells were sorted by

GFP into RLT lysis buffer (Qiagen) for qPCR or stained with propidium iodide (BD Bioscience) for cell death analysis. Samples were acquired using FACSAria III (BD Bioscience) and analyzed using FlowJo 10.7 software.

### PBMC reprogramming into iPSCs and iPSC quality control

PBMCs from individuals A-III-1 and A-II-3 (Fig. 1; Table 1) were reprogrammed into iPSCs using Sendai virus SeVdp(KOSM)302 L vector-170523-1 (Nishimura et al, 2017). Cells were cultured in RPMI 1640 GlutaMAX medium (61870-010, Gibco) containing 2 mM L-glutamine at 1 million cells per well of a six-well ultra-low attachment plate (3471, Corning). Culture medium was supplemented with 10% (v/v) fetal bovine serum (FBS, 10270106, Gibco) and 1% (v/v) penicillin-streptomycin (15070063, Thermo Fisher). After overnight culture, PBMCs were infected with the KOSM302L vector (harboring OCT4, SOX2, KLF4, and c-MYC genes) at a multiplicity of infection of 2 for 2 h. The medium was then changed to Essential 6 medium (A1516401, Gibco), and cells were transferred to Matrigel-coated plates (BDAA356231, Corning) and medium refreshed daily. From day 8, cells were cultured in Essential 8 medium (A1517001, Gibco), changed every other day. Emerging iPSC colonies were manually picked. The removal of the transgene vector was confirmed by reverse-transcriptase PCR (Appendix Table S7). Embryoid body aggregates were formed by detaching iPSC colonies at 60–70% confluence with 0.5 mM EDTA (15575020, Life Technologies) and culturing in suspension (100 rpm) in ultra-low-attachment plates (38071, Stemcell Technologies) in E8 medium with 10 μM ROCK inhibitor (Y-27632, 72304, Stemcell Technologies). The next day, medium was switched to DMEM/F-12 GlutaMAX medium (31331028, Gibco), 10% (v/v) knockout serum replacement (10828010, Life Technologies), 1% (v/v) MEM non-essential amino acids (11140050, Gibco), 0.1 mM β-mercaptoethanol (31350010, Gibco), and 1% penicillin-streptomycin, refreshed every 2nd day for 1 week. Embryoid bodies (15–20 per well) were plated on Matrigel-coated culture slides (734-0402, Falcon) for 2 weeks, with medium refreshed every other day, and fixed in 4% PFA for immunostaining (Appendix Table S8). For karyotyping, iPSCs from a 3.5 cm dish at 80% confluency were washed with PBS, detached with TrypLE Express (12604013, Life Technologies), collected in E8 medium, and centrifuged for 3 min at 500×g. The supernatant was removed, and cell pellets were stored at −80 °C. Karyotype was assessed by the Karyostat assay (Thermo Fisher).

### Genome editing of iPSCs

CRISPR/Cas9 was used to correct the R6C insulin mutation in patient iPSC lines ULBi011.5 (heterozygous) and ULBi012.31 (homozygous) using homology-directed repair (HDR). An isogenic control for patient iPSCs was generated using a 20-base guide (Appendix Table S3). The correction asymmetric single-stranded HDR donor was a 128-bp PCR amplicon using 91-base forward and 36-base reverse primers (Appendix Table S3). Templates were designed to disturb the protospacer adjacent motif (PAM) sequence and introduce a silent mutation to generate a restriction site of BtsI to facilitate the screening of clones using restriction enzymes. Ribonucleoprotein components were purchased from Integrated DNA Technologies, including CRISPR/Cas9 (1081061), crRNA (custom design from Benchling (Benchling [Biology Software], 2025)), tracrRNA (1075928), and the donor HDR template DNA

(custom design from Benchling). Two million cells were electroporated with ribonucleoprotein complex (tracrRNA:crRNA:Cas9 = 1:1:1.36) and 4 µg HDR template using Neon Transfection System (1100 V, 20 ms, 2 pulses, Thermo Fisher Scientific) and single-cell-cloned using low density seeding. Single clones were screened using PCR, followed by BtsI-v2 enzyme restriction (R0667S, New England) according to the manufacturer's protocol. Positive clones and isogenic controls were confirmed by Sanger sequencing. The top five off-target hits predicted by the online tool CRISPOR (Concordet and Haeussler, 2018) were checked, and no off-target indels were found (Appendix Table S3). CRISPRed iPSC lines were quality controlled as described above.

### Sanger sequencing

The R6C mutation in the proband and father of the proband in the first family (Fig. 1A) was identified by targeted Sanger sequencing and classified using ACMG/AMP and ACGS 2020_V4.01 guidelines. For iPSC lines, genomic DNA was extracted using QIAamp DNA Micro Kit (51306, Qiagen) and the R6C mutation confirmed by targeted Sanger sequencing (Microsynth, Germany) of PCR amplicons generated using iProof High-Fidelity DNA Polymerase (1725301, Bio-Rad). Primer sequences are shown in Appendix Table S3.

### iPSC culture and differentiation into β cells

The following iPSC lines were used: heterozygous ULBi011.5 (from individual A-II-3) and CRISPRed non-edited ULBi011.5-CR29; homozygous ULBi012.8 and ULBi012.31 (from individual A-III-1) and CRISPRed non-edited ULBi012.31-CR4; corrected ULBi012.31-CR3, ULBi012.31-CR41, ULBi011.5-CR21 and ULBi011.5-CR30; healthy control 1.023 (Fantuzzi et al, 2022). iPSCs were cultured in Matrigel-coated plates in E8 medium as described (Cosentino et al, 2018). iPSCs were differentiated into β cells following a previously described seven-stage protocol (Fig. EV3A, (Cosentino et al, 2018; Fantuzzi et al, 2022; Virgilio et al, 2025)). Cells were plated at a density of 2–2.3 million cells per 3.5 cm well in E8 medium with 10 µM ROCK inhibitor. After 24 h, differentiation was initiated. Up to the pancreatic progenitor stage (stage 4), cells were cultured in Matrigel-coated wells. At the end of this stage, cells were transferred into microwell plates at 850 cells per microwell (AggreWell400, StemCell Technologies) to form islet-like aggregates. Differentiation continued in microwells to the end of stage 7. In the long-culture differentiation protocol, aggregates were shifted to suspension culture (100 rpm) at the end of stage 5 and maintained in stage 7 medium (Barsby et al, 2022) for 4 weeks beyond stage 7 (Fig. EV3A, (Virgilio et al, 2025)).

### Transplantation of iPSC-β cells

Rag2 knockout mice (B6.Cg-Rag2tm1.1Cgn/J, The Jackson Laboratory) from SCANBUR were housed in the animal facility on a 12-h light/12-h dark cycle with ad libitum food. Transplantations were performed in 8 to 12-week-old male mice as described previously (De Franco et al, 2020; Virgilio et al, 2025). Around 1500 MACS-purified stage 7 iPSC-β cell aggregates were transplanted under the left kidney capsule, using a catheter and syringe or by inserting aggregates clotted with a blood drop from the recipient mouse. Mice were anaesthetized by IP injection of ketamine (100 mg/kg Nimatek, Dechra, UK) and xylazine (5 mg/kg Rompun, Bayer, Germany). Paracetamol was added to drinking water for 10 days

after surgery. IpGTTs were performed following a 6-h fast. Glycemia was measured using a glucometer (Accu-Chek Aviva Nano, Roche) at 0,15, 30, 60, 90, and 120 min after glucose injection (2 mg/g body weight). Mouse blood was sampled at 0, 15, 30, 60, and 90 min and plasma separated by centrifugation at 3000 RPM for 20 min at 4 °C for C-peptide quantification by human ultrasensitive C-peptide ELISA (10-1141-01, Mercodia). Dulaglutide (Trulicity, Eli Lilly) treatment was given after 4 months of transplantation at 1 mg/kg (dilution in saline) injected IP twice weekly for 2 months, with treatment washout in the last week.

### Western blot

iPSC-islets were lysed in Laemmli 2X buffer containing no reducing agents, sonicated (duty cycle 50% of 60 s), heated to 95 °C for 10 min, and stored at −80 °C. Samples were heated to 95 °C in gel sample buffer containing no reducing agents (nonreducing), or 200 mM DTT for 5 min (reducing), electrophoretically resolved on 12 or 4–12% NuPAGE gels. The gels were either untreated or treated with 100 mM DTT at 60 °C for 10 min, followed by electrotransfer to nitrocellulose. Primary antibodies were incubated at 4 °C overnight. Development of immunoblots used enhanced chemiluminescence, captured with a Bio-Rad ChemiDoc and quantified using Fiji. Antibodies are listed in Appendix Table S8.

### Min6 cells

Min6 (mouse insulinoma) cells (from Dr. D. Stoffers, University of Pennsylvania) were cultured in DMEM supplemented with 10% fetal bovine serum (FBS),100 IU/mL penicillin and 100 µg/mL streptomycin, and 0.05 mM β-mercaptoethanol.

### (Pro)insulin content and secretion

For static secretion experiments, 50 iPSC-islets were washed twice with glucose-free Krebs buffer and then preincubated for 90 min in 500 µL of βKrebs buffer (BK-100, Human Cell Design) containing 2.8 mM glucose. iPSC-islets were then sequentially incubated for 30-min intervals in βKrebs buffer with 2.8 mM, 16.8 mM glucose, and 16.8 mM glucose plus 10 µM forskolin (F6886, Sigma-Aldrich). Supernatants were collected and stored at −80 °C. Dynamic insulin secretion studies were performed on a PERI5LM perifusion system (BioRep, Miami, USA) with 2.8 mM, 16.8 mM glucose, 16.8 mM glucose + 50 ng/mL exendin-4 (E7144, Sigma-Aldrich), and 2.8 mM glucose + 30 mM KCl at a flow rate of 100 µL/min. Perifusate was stored at −20 °C. iPSC-islets were lysed by sonication (duty cycle 50% of 120 s) and extracted with acid-ethanol. Human insulin and proinsulin were assayed by ELISA (10-1113-10 and 10-1118-01, respectively, Mercodia). Secretion and content were normalized to iPSC-islet protein content determined by Bradford assay (5000006, Bio-Rad).

### Mitochondrial respiration

Oxygen consumption rate was measured using XFp Extracellular Flux Analyzer with consumables from Agilent. Stage 7 iPSC-islets were dispersed into single cells and seeded in Seahorse XFp Cell Culture Miniplates at 50,000 cells per well in long-culture differentiation medium. After overnight culture, the medium was refreshed to Ham's F-10 medium (41550, Gibco) supplemented with 0.75% fatty acid-free BSA, 0.5% (v/v) GlutaMAX and 1% penicillin-streptomycin, and cells were cultured for 1–2 days. On the test day, cells were preincubated in an assay buffer

supplemented with 0.2% (w/v) BSA and 0 mM glucose at pH 7.4 (Malmgren et al, 2009) for 1 h at 37 °C in a non-CO$_2$ incubator. Mitochondrial respiration was assessed at baseline, followed by sequential injections to achieve final concentrations of pyruvate (10 mM), oligomycin (5 µM), FCCP (4 µM), and a combination of rotenone and antimycin A (1 µM) in BSA-free assay buffer. Oxygen consumption rate was normalized to protein content. Basal respiration was calculated by subtracting nonmitochondrial respiration from the last measurement before oligomycin injection, ATP production by subtracting the minimum measurement after oligomycin injection from the last baseline measurement, and maximal respiration by subtracting nonmitochondrial respiration from the maximum measurement after FCCP injection.

### In vitro iPSC-islet treatment

Stage 7 iPSC-islets were exposed to 50 nM exendin-4, 50 nM dulaglutide (Trulicity, Eli Lilly), 50 nM liraglutide (Victoza, Novo Nordisk), 1 µM thapsigargin (T9033, Sigma-Aldrich), 5 µg/mL tunicamycin (T7765, Sigma-Aldrich), and 0.025 µg/mL brefeldin A (B6542, Sigma-Aldrich) in Ham's F-10 medium supplemented with 0.75% fatty acid-free BSA, 0.5% GlutaMAX and 1% penicillin-streptomycin. Vehicle PBS (for exendin-4, dulaglutide and liraglutide) or DMSO (for tunicamycin, thapsigargin, and brefeldin A) was added to the control condition in all experiments. Medium was refreshed every 24 h.

### Assessment of apoptosis

The proportion of viable and apoptotic cells was assessed using fluorescence microscopy following a 15-min incubation with DNA-binding dyes propidium iodide (5 µg/mL, P4170, Sigma-Aldrich) and Hoechst 33342 (10 µg/mL, B2261, Sigma-Aldrich). For each condition, at least ten iPSC-islets were counted by two observers, one of whom was blinded to the sample identities. Inter-observer agreement was >90%.

### Magnetic-activated cell sorting

Stage 7 iPSC-islets were washed twice with 0.5 mM EDTA and dissociated using Accumax (A7089, Sigma-Aldrich) incubation and mixing for 10 min at 37 °C on a MACSmix Tube Rotator (Miltenyi Biotec). Half volume of KnockOut Serum Replacement was added and gently pipetted, and 10 µM ROCK inhibitor was added to the cell solution. For labeling, 10 million cells were resuspended in 300 µL MACS buffer (PBS, pH 7.2, 0.5% BSA, 2 mM EDTA) plus 200 µL CD49a antibody (PE-conjugated mouse anti-human CD49a, clone SR84, 559596, BD Bioscience) (Veres et al, 2019). Cells were incubated in the dark for 21 min at room temperature on MACSmix Tube Rotator, washed twice with 15 mL MACS buffer, resuspended in 300 µL MACS buffer with 20 µl anti-PE MicroBeads (130-048-801, Miltenyi Biotec) per 10 million cells for 15 min at 4 °C. After a final wash in MACS buffer, cells were sorted using MACS magnetic separators and columns (130-042-201, Miltenyi Biotec). The CD49a-positive fraction was seeded at one million cells per AggreWell.

### Conditioned media test in HepG2 cells

HepG2 cells were seeded in 12-well plates pre-coated with collagen I Rat Tail (1:100 diluted in PBS, Gibco, A10483-01) and cultured using DMEM-GlutaMAX (Gibco,61965-026) supplemented with 10% FBS (Eurobio) and 1% penicillin-streptomycin (Gibco). Conditioned medium was collected after 48-h of long-cultured iPSC-islet culture in Ham's F-10 medium supplemented with 0.75% fatty acid-free BSA, 0.5% GlutaMAX, and 1% penicillin-streptomycin. HepG2 cells were serum starved for 4 days before incubation with conditioned media at a volume corresponding to 5 ng insulin. Medium only was used as a negative control and 5 ng NovoRapid insulin (Novo Nordisk) as a positive control. Cells were treated for 30 min, lysed in RIPA buffer (Sigma) with anti-protease cOmplete and PhosSTOP tablets (Roche) and sonicated. About 15 µg protein was resolved in 4–12% Bis-Tris gel and transferred to PVDF membrane using iBLOT Dry Blotting System (Thermo Fisher Scientific). Membranes were incubated with anti-p-AKT, AKT, or β-Actin antibodies. Species-specific HRP-linked secondary antibodies (Cell Signaling Technology) were utilized. Visualization was carried out on an iBright imager (Invitrogen) following exposure to ECL (GE Healthcare). Image quantification was carried out on Fiji. Antibodies are listed in Appendix Table S8.

### RNA extraction, qPCR, and RNA sequencing

Poly(A)$^+$-RNA was isolated from iPSC-islets using Dynabeads mRNA DIRECT Purification kit (61012, Invitrogen). Reverse transcription was done using Reverse Transcriptase Core Kit (RT-RTCK-03, Eurogentec). Real-time qPCR amplification was performed using SsoAdvanced Universal SYBR Green Supermix (1725274, Bio-Rad) on the CFX Connect instrument (Bio-Rad) and compared to a standard curve. Expression values were normalized to reference genes β-actin (ACTB) and vesicle-associated membrane protein-associated protein A (VAPA) (Alvelos et al, 2021). Primers are listed in Appendix Table S7.

Total RNA from MACS-purified iPSC-β cell aggregates was isolated using the RNeasy Plus Micro kit (74034, Qiagen). RNA concentration and integrity were measured using Bioanalyzer RNA 6000 Nano assay (5067-1511, Agilent) on the Agilent 2100 Bioanalyzer. Libraries were prepared using NEBNext Ultra II Directional RNA Library Prep Kit for Illumina (E7760L, New England BioLabs) and Novogene NGS Stranded RNA Library Prep Set (PT044, Novogene). All libraries were paired-end sequenced (2x151bp) at 200 million reads on the Illumina NovaSeq 6000 and NovaSeq X plus instrument (Illumina).

For EndoC-βH1 cells, the RNeasy Micro kit (74004, Qiagen) was used to extract total RNA. cDNA was synthesized using the First-Strand cDNA kit (K1612, Thermo Fisher Scientific). RT-qPCR was executed using Power SYBR Green mix (4368708, Invitrogen) with a QuantStudio 3 analyzer. Primers were designed using primer blast and synthesized (Eurofins Genomics). The sequences of the primers are listed in Appendix Table S7.

### Immunocytochemistry of iPSC-β cell aggregates

For single-cell staining, cells were dissociated and seeded in culture medium containing 10 µM ROCK inhibitor onto Matrigel-coated ICC chambers. After 24 h, cells were washed once with PBS, fixed in 4% PFA for 15 min, permeabilized with 0.25% Triton X-100 in PBS for 10 min, washed again with PBS, and blocked for 30 min at room temperature using UltraCruz Blocking Reagent (Santa Cruz Biotechnology, Cat#SC-516214) or 3% BSA. Primary antibodies diluted in 0.1% Tween in PBS or 3% BSA were incubated overnight at 4 °C, and secondary antibodies for 45 min at room temperature. Samples were mounted using Vectashield with DAPI (VEC.H-1200, Vector Laboratories) or stained with DAPI (D1306, Thermo Fisher)

at 1 mg/mL, mounted with Glycergel mounting medium (C0563, Dako) and covered with glass coverslips. Image quantifications were performed using custom CellProfiler 4 pipelines (Stirling et al, 2021). Antibodies are listed in Appendix Table S8. Immunofluorescence pictures were taken on an Axio Imager A1 (3521000414, Carl Zeiss) or Axio Observer 7 inverted microscope (4919170001000, Carl Zeiss).

### Statistical analysis

Statistical analyses were performed with GraphPad Prism (version 10, GraphPad Software). Data points represent independent experiments, and different symbols represent different iPSC lines in each category (homozygous, heterozygous, or isogenic corrected cells). In box plots, the median is shown by a horizontal line; $25^{th}$ and $75^{th}$ percentiles are at the bottom and top of the boxes; whiskers represent the minimum and maximum values. In time course experiments (perifusion, Seahorse, and IpGTT), data are shown as mean ± s.e.m. Sample sizes were estimated using power calculations with a two-tailed significance threshold of 0.05 and power set at 0.8, to ensure sensitivity to detect expected effect sizes. Student's t-test was used to assess significance when comparing two groups with normal distribution (by Shapiro–Wilk test); otherwise, a nonparametric test was applied. For experiments with >2 groups, paired or unpaired one-way or two-way ANOVA or mixed-model analysis (in case of a missing value) was followed by two-tailed t-tests with the indicated correction for multiple comparisons in each figure panel.

For RNA sequencing analyses, FastQC version 0.12.1 (Andrews, 2023) was used to quality control raw reads, and fastp version 0.23.2 (Chen et al, 2018) to filter low-quality reads and adapter sequences. Salmon version 1.10.3 (Patro et al, 2017) was used to quantify filtered raw reads using pseudo-alignment on the transcriptome reference derived from GRCh38 primary assembly (Genecode version 46), resulting in an average mapping rate of 88%. Differential gene expression analysis was conducted using tximeta version 1.22.1 (Love et al, 2020) and DESeq2 version 1.44.0 (Love et al, 2014). Gene set enrichment assay was conducted using fgsea version 1.33.1 (Korotkevich et al, 2021). Gene-wise Z-scores were calculated from TPM derived from Salmon and imported via tximeta to stabilize scale differences in RNA-seq data. For each gene, the mean is subtracted and divided by the standard deviation across samples, centering and scaling expression values to enable cross-sample comparison.

### Graphics

The synopsis image, Figs. 1, 6A, 7A and EV1D, EV3A; Appendix Fig. S1 were created with BioRender.com.

### Study approval

Informed consent was obtained from all human subjects (the patients and/or their guardians). The experiments conformed to the principles set out in the WMA Declaration of Helsinki and the Department of Health and Human Services Belmont Report. Human iPSCs were differentiated into β cells with approval of the Ethical Committee of Erasmus Hospital, Université Libre de Bruxelles (P2019/498, A2024/211). Mouse experiments were approved by the Commission d'Ethique et du Bien Être Animal, Faculty of Medicine, Université Libre de Bruxelles, following the European Convention for the Protection of Vertebrate Animals used for Experimental and other Scientific Purposes (European Treaty Series No. 123). The UK Biobank analysis was conducted under UK Biobank project number 103356. All UK participants in this study gave informed consent for genetic studies and were approved by the North Wales ethics committee (no. 17/WA/0327).

## Data availability

RNA-Seq data: Gene Expression Omnibus GSE300608 (https://www.ncbi.nlm.nih.gov/geo/query/acc.cgi?acc=GSE300608).

The source data of this paper are collected in the following database record: biostudies:S-SCDT-10_1038-S44321-025-00362-9.

## Peer review information

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

---

### The Paper Explained

#### Problem

Diabetes can sometimes be caused by a "single letter typo" (mutation) in the DNA, called monogenic diabetes. One of these forms is called insulin R6C, where the $6^{th}$ amino acid arginine in the insulin precursor protein is changed to a cysteine. It affects how the blood sugar-controlling hormone insulin is produced. It was unclear if having one copy of this typo (inherited from one parent) is enough to cause the disease, or if two copies (inherited from both parents) are necessary.

#### Results

In this study, we used patient data, genetic analyses, and stem cells to understand the impact of the R6C mutation in insulin. We found that individuals with two R6C copies develop diabetes in childhood. Those with only one copy are not, however, at increased risk of developing diabetes. To explore this further, we created insulin-producing β cells from R6C patients' stem cells. β Cells with two R6C copies produced and released less insulin, also when these β cells were transplanted into mice. In contrast, β cells with one R6C copy performed nearly as well as normal β cells.

#### Impact

Overall, our results show that the R6C mutation causes diabetes only when both copies of the DNA are affected. People carrying just one copy remain healthy unless exposed to other diabetes risk factors. Our findings highlight the value of patients' stem-cell models to study genetic mutations and test treatments, helping doctors provide better guidance to families carrying these rare genetic changes.

Albagli-Curiel O, Lécluse Y, Pognonec P, Boulukos KE, Martin P (2007) A new generation of pPRIG-based retroviral vectors. BMC Biotechnol 7:85

ALFA (Allele Frequency Aggregator) (2024) https://www.ncbi.nlm.nih.gov/snp/docs/gsr/alfa/ [DATASET]

Alfaris N, Waldrop S, Johnson V, Boaventura B, Kendrick K, Stanford FC (2024) GLP-1 single, dual, and triple receptor agonists for treating type 2 diabetes and obesity: a narrative review. EClinicalMedicine 75:102782

All of Us Research Program Investigators, Denny JC, Rutter JL, Goldstein DB, Philippakis A, Smoller JW, Jenkins G, Dishman E (2019) The 'All of Us' Research Program. N Engl J Med 381:668–676

Alvelos MI, Szymczak F, Castela Â, Marín-Cañas S, de Souza BM, Gkantounas I, Colli M, Fantuzzi F, Cosentino C, Igoillo-Esteve M et al (2021) A functional genomic approach to identify reference genes for human pancreatic beta cell real-time quantitative RT-PCR analysis. Islets 13:51–65

American Diabetes Association Professional Practice C (2025) 2. Diagnosis and classification of diabetes: standards of care in diabetes-2025. Diabetes Care 48:S27–S49

Andrews S (2023) Babraham Bioinformatics - FastQC.

Annesley SJ, Lay ST, De Piazza SW, Sanislav O, Hammersley E, Allan CY, Francione LM, Bui MQ, Chen Z-P, Ngoei KRW et al (2016) Immortalized Parkinson's disease lymphocytes have enhanced mitochondrial respiratory activity. Dis Model Mech 9:1295–1305

Ashburner M, Ball CA, Blake JA, Botstein D, Butler H, Cherry JM, Davis AP, Dolinski K, Dwight SS, Eppig JT et al (2000) Gene ontology: tool for the unification of biology. The Gene Ontology Consortium. Nat Genet 25:25–29

Atkinson EG, Artomov M, Loboda AA, Rehm HL, MacArthur DG, Karczewski KJ, Neale BM, Daly MJ (2023) Discordant calls across genotype discovery approaches elucidate variants with systematic errors. Genome Res 33:999–1005

Balboa D, Barsby T, Lithovius V, Saarimäki-Vire J, Omar-Hmeadi M, Dyachok O, Montaser H, Lund P-E, Yang M, Ibrahim H et al (2022) Functional, metabolic and transcriptional maturation of human pancreatic islets derived from stem cells. Nat Biotechnol 40:1042–1055

Balboa D, Saarimäki-Vire J, Borshagovski D, Survila M, Lindholm P, Galli E, Eurola S, Ustinov J, Grym H, Huopio H et al (2018) Insulin mutations impair beta-cell development in a patient-derived iPSC model of neonatal diabetes. Elife 7:e38519

Bansal V, Gassenhuber J, Phillips T, Oliveira G, Harbaugh R, Villarasa N, Topol EJ, Seufferlein T, Boehm BO (2017) Spectrum of mutations in monogenic diabetes genes identified from high-throughput DNA sequencing of 6888 individuals. BMC Med 15:213

Barsby T, Ibrahim H, Lithovius V, Montaser H, Balboa D, Vähäkangas E, Chandra V, Saarimäki-Vire J, Otonkoski T (2022) Differentiating functional human islet-like aggregates from pluripotent stem cells. STAR Protoc 3:101711

Benchling [Biology Software] (2025)

Boesgaard TW, Pruhova S, Andersson EA, Cinek O, Obermannova B, Lauenborg J, Damm P, Bergholdt R, Pociot F, Pisinger C et al (2010) Further evidence that mutations in INScan be a rare cause of Maturity-Onset Diabetes of the Young (MODY). BMC Med Genet 11:42

Bonnefond A, Semple RK (2022) Achievements, prospects and challenges in precision care for monogenic insulin-deficient and insulin-resistant diabetes. Diabetologia 65:1782–1795

Carey DJ, Fetterolf SN, Davis FD, Faucett WA, Kirchner HL, Mirshahi U, Murray MF, Smelser DT, Gerhard GS, Ledbetter DH (2016) The Geisinger MyCode Community Health Initiative: an electronic health record-linked biobank for precision medicine research. Genet Med 18:906–913

Chen S, Zhou Y, Chen Y, Gu J (2018) fastp: an ultra-fast all-in-one FASTQ preprocessor. Bioinformatics 34:i884–i890

Chua C, Tan CSH, Lim SC, Vasanwala RF (2024) A unique phenotype of maturity-onset diabetes of the young with a novel disease-causing insulin gene variant. JCEM Case Rep 3:luae230

Colombo C, Porzio O, Liu M, Massa O, Vasta M, Salardi S, Beccaria L, Monciotti C, Toni S, Pedersen O et al (2008) Seven mutations in the human insulin gene linked to permanent neonatal/infancy-onset diabetes mellitus. J Clin Invest 118:2148–2156

Concordet J-P, Haeussler M (2018) CRISPOR: intuitive guide selection for CRISPR/Cas9 genome editing experiments and screens. Nucleic Acids Res 46:W242–W245

Cosentino C, Toivonen S, Diaz Villamil E, Atta M, Ravanat J-L, Demine S, Schiavo AA, Pachera N, Deglasse J-P, Jonas J-C et al (2018) Pancreatic β-cell tRNA hypomethylation and fragmentation link TRMT10A deficiency with diabetes. Nucleic Acids Res 46:10302–10318

De Franco E, Lytrivi M, Ibrahim H, Montaser H, Wakeling MN, Fantuzzi F, Patel K, Demarez C, Cai Y, Igoillo-Esteve M et al (2020) YIPF5 mutations cause neonatal diabetes and microcephaly through endoplasmic reticulum stress. J Clin Invest 130:6338–6353

De Franco E, Owens NDL, Montaser H, Wakeling MN, Saarimaki-Vire J, Triantou A, Ibrahim H, Balboa D, Caswell RC, Jennings RE et al (2023) Primate-specific ZNF808 is essential for pancreatic development in humans. Nat Genet 55:2075–2081

Drucker DJ (2018) Mechanisms of action and therapeutic application of glucagon-like peptide-1. Cell Metab 27:740–756

Edghill EL, Flanagan SE, Patch A-M, Boustred C, Parrish A, Shields B, Shepherd MH, Hussain K, Kapoor RR, Malecki M et al (2008) Insulin mutation screening in 1,044 patients with diabetes: mutations in the INS gene are a common cause of neonatal diabetes but a rare cause of diabetes diagnosed in childhood or adulthood. Diabetes 57:1034–1042

Fabregat A, Jupe S, Matthews L, Sidiropoulos K, Gillespie M, Garapati P, Haw R, Jassal B, Korninger F, May B et al (2018) The Reactome pathway knowledgebase. Nucleic Acids Res 46:D649–D655

Fantuzzi F, Toivonen S, Schiavo AA, Chae H, Tariq M, Sawatani T, Pachera N, Cai Y, Vinci C, Virgilio E et al (2022) In depth functional characterization of human induced pluripotent stem cell-derived beta cells in vitro and in vivo. Front Cell Dev Biol 10:967765

Faul F, Erdfelder E, Buchner A, Lang A-G (2009) Statistical power analyses using G*Power 3.1: Tests for correlation and regression analyses. Behavior Research Methods 41:1149–1160

Garin I, Edghill EL, Akerman I, Rubio-Cabezas O, Rica I, Locke JM, Maestro MA, Alshaikh A, Bundak R, del Castillo G et al (2010) Recessive mutations in the INS gene result in neonatal diabetes through reduced insulin biosynthesis. Proc Natl Acad Sci USA 107:3105–3110

George MN, Leavens KF, Gadue P (2021) Genome editing human pluripotent stem cells to model β-cell disease and unmask novel genetic modifiers. Front Endocrinol 12:682625

Goodrich JK, Singer-Berk M, Son R, Sveden A, Wood J, England E, Cole JB, Weisburd B, Watts N, Caulkins L et al (2021) Determinants of penetrance and variable expressivity in monogenic metabolic conditions across 77,184 exomes. Nat Commun 12:3505

Gorgogietas V, Rajaei B, Heeyoung C, Santacreu BJ, Marín-Cañas S, Salpea P, Sawatani T, Musuaya A, Arroyo MN, Moreno-Castro C et al (2023) GLP-1R agonists demonstrate potential to treat Wolfram syndrome in human preclinical models. Diabetologia 66:1306–1321

Greeley SAW, Polak M, Njolstad PR, Barbetti F, Williams R, Castano L, Raile K, Chi DV, Habeb A, Hattersley AT et al (2022) ISPAD Clinical Practice Consensus Guidelines 2022: The diagnosis and management of monogenic diabetes in children and adolescents. Pediatr Diabetes 23:1188–1211

Guo H, Sun J, Li X, Xiong Y, Wang H, Shu H, Zhu R, Liu Q, Huang Y, Madley R et al (2018) Positive charge in the n-region of the signal peptide contributes to efficient post-translational translocation of small secretory preproteins. J Biol Chem 293:1899–1907

Guo H, Xiong Y, Witkowski P, Cui J, Wang L, Sun J, Lara-Lemus R, Haataja L, Hutchison K, Shan S et al (2014) Inefficient translocation of preproinsulin contributes to pancreatic β cell failure and late-onset diabetes. J Biol Chem 289:16290–16302

Gutierrez Guarnizo SA, Kellogg MK, Miller SC, Tikhonova EB, Karamysheva ZN, Karamyshev AL (2023) Pathogenic signal peptide variants in the human genome. NAR Genom Bioinform 5:lqad093

Haßdenteufel S, Johnson N, Paton AW, Paton JC, High S, Zimmermann R (2018) Chaperone-mediated Sec61 channel gating during ER import of small precursor proteins overcomes Sec61 inhibitor-reinforced energy barrier. Cell Rep 23:1373–1386

Huerta-Chagoya A, Schroeder P, Mandla R, Li J, Morris L, Vora M, Alkanaq A, Nagy D, Szczerbinski L, Madsen JGS et al (2024) Rare variant analyses in 51,256 type 2 diabetes cases and 370,487 controls reveal the pathogenicity spectrum of monogenic diabetes genes. Nat Genet 56:2370–2379

Hussain S, Ali JM, Jalaludin MY, Harun F (2013) Permanent neonatal diabetes due to a novel insulin signal peptide mutation. Pediatric Diabetes 14:299–303

International Diabetes Federation (2025) IDF diabetes atlas 11th edn. International Diabetes Federation, Brussels

Iwasa Y, Miyata S, Tomita T, Yokota N, Miyauchi M, Mori R, Matsushita S, Suzuki R, Saeki Y, Kawahara H (2025) TanGIBLE: a selective probe for evaluating hydrophobicity-exposed defective proteins in live cells. J Cell Biol 224:e202109010

James NM, Stanford KI (2025) Obesity and exercise: new insights and perspectives. Endocr Rev 46:763–789

Kanehisa M, Goto S (2000) KEGG: kyoto encyclopedia of genes and genomes. Nucleic Acids Res 28:27–30

Karczewski KJ, Francioli LC, Tiao G, Cummings BB, Alföldi J, Wang Q, Collins RL, Laricchia KM, Ganna A, Birnbaum DP et al (2020) The mutational constraint spectrum quantified from variation in 141,456 humans. Nature 581:434–443

Kim S-H, Ma X, Klupa T, Powers C, Pezzolesi M, Warram JH, Rich SS, Krolewski AS, Doria A (2003) Genetic modifiers of the age at diagnosis of diabetes (MODY3) in carriers of hepatocyte nuclear factor-1alpha mutations map to chromosomes 5p15, 9q22, and 14q24. Diabetes 52:2182–2186

Kingdom R, Beaumont RN, Wood AR, Weedon MN, Wright CF (2024) Genetic modifiers of rare variants in monogenic developmental disorder loci. Nat Genet 56:861–868

Korotkevich G, Sukhov V, Budin N, Shpak B, Artyomov MN, Sergushichev A (2021) Fast gene set enrichment analysis. Preprint at *bioRxiv* https://doi.org/10.1101/060012

Kristinsson SY, Thorolfsdottir ET, Talseth B, Steingrimsson E, Thorsson AV, Helgason T, Hreidarsson AB, Arngrimsson R (2001) MODY in Iceland is associated with mutations in HNF-1alpha and a novel mutation in NeuroD1. Diabetologia 44:2098–2103

Lang S, Pfeffer S, Lee P-H, Cavalié A, Helms V, Förster F, Zimmermann R (2017) An update on Sec61 channel functions, mechanisms, and related diseases. Front Physiol 8:887

Laver TW, Franco ED, Johnson MB, Patel KA, Ellard S, Weedon MN, Flanagan SE, Wakeling MN (2022) SavvyCNV: genome-wide CNV calling from off-target reads. PLoS Comput Biol 18:e1009940

Leech J, Beaumont RN, Arni AM, Chundru VK, Sharp LN, Colclough K, Hattersley AT, Weedon MN, Patel KA (2025) Common genetic variants modify disease risk and clinical presentation in monogenic diabetes. Nat Metab 7:1819–1829

Li M, Popovic N, Wang Y, Chen C, Polychronakos C (2023a) Incomplete penetrance and variable expressivity in monogenic diabetes; a challenge but also an opportunity. Rev Endocr Metab Disord 24:673–684

UK Biobank Whole-Genome Sequencing Consortium (2025) Whole-genome sequencing of 490,640 UK Biobank participants. 645:692-701

Li X, Itani OA, Haataja L, Dumas KJ, Yang J, Cha J, Flibotte S, Shih H-J, Delaney CE, Xu J et al (2019) Requirement for translocon-associated protein (TRAP) α in insulin biogenesis. Sci Adv 5:eaax0292

Li Y, Polychronakos C (2024) Parsing the spectrum of allelic architectures in diabetes. Nat Genet 56:2297–2298

Liberzon A, Birger C, Thorvaldsdóttir H, Ghandi M, Mesirov JP, Tamayo P (2015) The Molecular Signatures Database (MSigDB) hallmark gene set collection. Cell Syst 1:417–425

Liu M, Lara-Lemus R, Shan S, Wright J, Haataja L, Barbetti F, Guo H, Larkin D, Arvan P (2012) Impaired cleavage of preproinsulin signal peptide linked to autosomal-dominant diabetes. Diabetes 61:828–837

Liu M, Sun J, Cui J, Chen W, Guo H, Barbetti F, Arvan P (2015) INS-gene mutations: from genetics and beta cell biology to clinical disease. Mol Aspects Med 42:3–18

Liu X, Li C, Mou C, Dong Y, Tu Y (2020) dbNSFP v4: a comprehensive database of transcript-specific functional predictions and annotations for human nonsynonymous and splice-site SNVs. Genome Med 12:103

Love MI, Huber W, Anders S (2014) Moderated estimation of fold change and dispersion for RNA-seq data with DESeq2. Genome Biology 15: 550

Love MI, Soneson C, Hickey PF, Johnson LK, Pierce NT, Shepherd L, Morgan M, Patro R (2020) Tximeta: reference sequence checksums for provenance identification in RNA-seq. PLoS Comput Biol 16:e1007664

Luckett AM, Hawkes G, Green HD, De Franco E, Hagopian WA, Roep BO, Weedon MN, Oram RA, Johnson MB, EXE-T1D consortium (2025) Type 1 diabetes genetic risk contributes to phenotypic presentation in monogenic autoimmune diabetes. Diabetes 74:243–248

Malmgren S, Nicholls DG, Taneera J, Bacos K, Koeck T, Tamaddon A, Wibom R, Groop L, Ling C, Mulder H et al (2009) Tight coupling between glucose and mitochondrial metabolism in clonal beta-cells is required for robust insulin secretion. J Biol Chem 284:32395–32404

Meur G, Simon A, Harun N, Virally M, Dechaume A, Bonnefond A, Fetita S, Tarasov AI, Guillausseau P-J, Boesgaard TW et al (2010) Insulin gene mutations resulting in early-onset diabetes: marked differences in clinical presentation, metabolic status, and pathogenic effect through endoplasmic reticulum retention. Diabetes 59:653–661

Miller SC, Tikhonova EB, Hernandez SM, Dufour JM, Karamyshev AL (2024) Loss of preproinsulin interaction with signal recognition particle activates protein quality control, decreasing mRNA stability. J Mol Biol 436:168492

Mirshahi UL, Colclough K, Wright CF, Wood AR, Beaumont RN, Tyrrell J, Laver TW, Stahl R, Golden A, Goehringer JM et al (2022) Reduced penetrance of MODY-associated HNF1A/HNF4A variants but not GCK variants in clinically unselected cohorts. Am J Hum Genet 109:2018–2028

Mittendorfer B, Patterson BW, Haire-Joshu D, Cahill AG, Cade WT, Stein RI, Klein S (2023) Insulin sensitivity and β-cell function during early and late pregnancy in women with and without gestational diabetes mellitus. Diabetes Care 46:2147–2154

Molven A, Ringdal M, Nordbø AM, Ræder H, Støy J, Lipkind GM, Steiner DF, Philipson LH, Bergmann I, Aarskog D et al (2008) Mutations in the insulin gene can cause MODY and autoantibody-negative type 1 diabetes. Diabetes 57:1131–1135

Nesmeyanova MA, Karamyshev AL, Karamysheva ZN, Kalinin AE, Ksenzenko VN, Kajava AV (1997) Positively charged lysine at the N-terminus of the signal peptide of the *Escherichia coli* alkaline phosphatase provides the secretion

efficiency and is involved in the interaction with anionic phospholipids. FEBS Lett 403:203–207

Nishimura K, Ohtaka M, Takada H, Kurisaki A, Tran NVK, Tran YTH, Hisatake K, Sano M, Nakanishi M (2017) Simple and effective generation of transgene-free induced pluripotent stem cells using an auto-erasable Sendai virus vector responding to microRNA-302. Stem Cell Res 23:13–19

Oram RA, Patel K, Hill A, Shields B, McDonald TJ, Jones A, Hattersley AT, Weedon MN (2016) A type 1 diabetes genetic risk score can aid discrimination between type 1 and type 2 diabetes in young adults. Diabetes Care 39:337–344

Patro R, Duggal G, Love MI, Irizarry RA, Kingsford C (2017) Salmon provides fast and bias-aware quantification of transcript expression. Nat Methods 14:417–419

Perera LA, Hattersley AT, Harding HP, Wakeling MN, Flanagan SE, Mohsina I, Raza J, Gardham A, Ron D, De Franco E (2023) Infancy-onset diabetes caused by de-regulated AMPylation of the human endoplasmic reticulum chaperone BiP. EMBO Mol Med 15:e16491

Pettersen EF, Goddard TD, Huang CC, Meng EC, Couch GS, Croll TI, Morris JH, Ferrin TE (2021) UCSF ChimeraX: structure visualization for researchers, educators, and developers. Protein Sci 30:70–82

Ravassard P, Hazhouz Y, Pechberty S, Bricout-Neveu E, Armanet M, Czernichow P, Scharfmann R (2011) A genetically engineered human pancreatic β cell line exhibiting glucose-inducible insulin secretion. J Clin Invest 121:3589–3597

Rutkowski DT, Arnold SM, Miller CN, Wu J, Li J, Gunnison KM, Mori K, Sadighi Akha AA, Raden D, Kaufman RJ (2006) Adaptation to ER stress is mediated by differential stabilities of pro-survival and pro-apoptotic mRNAs and proteins. PLoS Biol 4:e374

Sánchez WN, Driessen AJM, Wilson CAM (2025) Protein targeting to the ER membrane: multiple pathways and shared machinery. Crit Rev Biochem Mol Biol 60:33–79

Schirinzi T, Salvatori I, Zenuni H, Grillo P, Valle C, Martella G, Mercuri NB, Ferri A (2022) Pattern of mitochondrial respiration in peripheral blood cells of patients with Parkinson's disease. Int J Mol Sci 23:10863

Soty M, Visa M, Soriano S, del Carmen Carmona M, Nadal Á, Novials A (2011) Involvement of ATP-sensitive potassium (KATP) channels in the loss of beta-cell function induced by human islet amyloid polypeptide. J Biol Chem 286:40857–40866

Stirling DR, Swain-Bowden MJ, Lucas AM, Carpenter AE, Cimini BA, Goodman A (2021) CellProfiler 4: improvements in speed, utility and usability. BMC Bioinformatics 22:433

Støy J, De Franco E, Ye H, Park S-Y, Bell GI, Hattersley AT (2021) In celebration of a century with insulin - Update of insulin gene mutations in diabetes. Mol Metab 52:101280

Støy J, Edghill EL, Flanagan SE, Ye H, Paz VP, Pluzhnikov A, Below JE, Hayes MG, Cox NJ, Lipkind GM et al (2007) Insulin gene mutations as a cause of permanent neonatal diabetes. Proc Natl Acad Sci USA 104:15040–15044

Subramanian A, Tamayo P, Mootha VK, Mukherjee S, Ebert BL, Gillette MA, Paulovich A, Pomeroy SL, Golub TR, Lander ES et al (2005) Gene set enrichment analysis: a knowledge-based approach for interpreting genome-wide expression profiles. Proc Natl Acad Sci USA 102:15545–15550

Sudlow C, Gallacher J, Allen N, Beral V, Burton P, Danesh J, Downey P, Elliott P, Green J, Landray M et al (2015) UK biobank: an open access resource for identifying the causes of a wide range of complex diseases of middle and old age. PLoS Med 12:e1001779

Sun KY, Bai X, Chen S, Bao S, Zhang C, Kapoor M, Backman J, Joseph T, Maxwell E, Mitra G et al (2024) A deep catalogue of protein-coding variation in 983,578 individuals. Nature 631:583–592

Szabat M, Page MM, Panzhinskiy E, Skovsø S, Mojibian M, Fernandez-Tajes J, Bruin JE, Bround MJ, Lee JTC, Xu EE et al (2016) Reduced insulin production

relieves endoplasmic reticulum stress and induces β cell proliferation. Cell Metab 23:179–193

Taliun D, Harris DN, Kessler MD, Carlson J, Szpiech ZA, Torres R, Taliun SAG, Corvelo A, Gogarten SM, Kang HM et al (2021) Sequencing of 53,831 diverse genomes from the NHLBI TOPMed program. Nature 590:290–299

Tans R, Glendorf T, van Herwaarden AE, Venselaar H, van Rijswijck DMH, Wevers RA, Gloerich J, van Gool A, Tack CJ (2024) A rare homozygous INS variant causes adult-onset diabetes. BMJ Open Diabetes Res Care 12:e004418

Veres A, Faust AL, Bushnell HL, Engquist EN, Kenty JH-R, Harb G, Poh Y-C, Sintov E, Gürtler M, Pagliuca FW et al (2019) Charting cellular identity during human in vitro β-cell differentiation. Nature 569:368–373

Virgilio E, Tielens S, Bonfield G, Nian F-S, Sawatani T, Vinci C, Govier M, Montaser H, Lartigue R, Arunagiri A et al (2025) Recessive *TMEM167A* variants cause neonatal diabetes, microcephaly and epilepsy syndrome. J Clin Invest 135:e195756

Wakeling MN (2018) SavvySuite.

Wang H, Saint-Martin C, Xu J, Ding L, Wang R, Feng W, Liu M, Shu H, Fan Z, Haataja L et al (2020) Biological behaviors of mutant proinsulin contribute to the phenotypic spectrum of diabetes associated with insulin gene mutations. Mol Cell Endocrinol 518:111025

Wang J, Takeuchi T, Tanaka S, Kubo SK, Kayo T, Lu D, Takata K, Koizumi A, Izumi T (1999) A mutation in the insulin 2 gene induces diabetes with severe pancreatic beta-cell dysfunction in the Mody mouse. J Clin Invest 103:27–37

Wang L, Pattnaik A, Sahoo SS, Stone EG, Zhuang Y, Benton A, Tajmul M, Chakravorty S, Dhawan D, Nguyen MA et al (2024) Unbiased discovery of cancer pathways and therapeutics using Pathway Ensemble Tool and Benchmark. Nat Commun 15:7288

Wright CF, Sharp LN, Jackson L, Murray A, Ware JS, MacArthur DG, Rehm HL, Patel KA, Weedon MN (2024) Guidance for estimating penetrance of monogenic disease-causing variants in population cohorts. Nat Genet 56:1772–1779

Xu X, Bell TW, Le T, Zhao I, Walker E, Wang Y, Xu N, Soleimanpour SA, Russ HA, Qi L et al (2024) Role of Sec61α2 translocon in insulin biosynthesis. Diabetes 73:2034–2044

Xu X, Huang Y, Li X, Arvan P, Liu M (2022) The Role of TRAPγ/SSR3 in preproinsulin translocation into the endoplasmic reticulum. Diabetes 71:440–452

## Acknowledgements

We thank Anyishaï Musuaya, Aurélien Augenlicht, Angéline Bilheu, Jessica Capitaine, Nayana Rodrigues Lé, Isabelle Millard, Naïma Belmahjoubi (ULB Center for Diabetes Research, Université Libre de Bruxelles), and Solja Eurola (Stem Cells and Metabolism Research Program, University of Helsinki) for excellent technical support. We thank Décio L. Eizirik for valuable discussions. This work is funded by the Innovative Medicines Initiative 2 Joint Undertaking under grant agreement No 115797 (INNODIA) and 945268 (INNODIA HARVEST), receiving support from the Union's Horizon 2020 research and innovation program and European Federation of Pharmaceutical Industries and Associations (EFPIA), Breakthrough T1D and the Leona M. and Harry B. Helmsley Charitable Trust, the European Union Horizon Health project NEMESIS, the Belgian Fonds National de la Recherche Scientifique (FNRS), FNRS-Weave, Walloon Region strategic axis Fonds de la Recherche Scientifique (FRFS)–Walloon Excellence in Life Sciences and Biotechnology (WELBIO), Win4Excellence GT4Health Research Foundation Flanders (FWO) & Fund for Scientific Research (FRS)-FNRS Excellence of Science (EOS) project Pandarome, the National Institutes of Health funding NIH-R01DK48280 and NIH-R01DK111174, and the National Institute for Health Research (NIHR) Exeter Biomedical Research Centre, and Clinical Research facilities, Exeter, UK. YT and FNS are FNRS-Fund for Research Training in Industry and Agriculture (FRIA)

fellows. KAP is funded by the Wellcome Trust (219606/Z/19/Z). The Wellcome Trust, MRC, and NIHR had no role in the design and conduct of the study; collection, management, analysis, and interpretation of the data; preparation, review, or approval of the manuscript; and decision to submit the manuscript for publication. The views expressed are those of the author(s) and not necessarily those of the Wellcome Trust, Department of Health, NHS or NIHR.

## Author contributions

**Yue Tong**: Conceptualization; Data curation; Software; Formal analysis; Validation; Investigation; Visualization; Methodology; Writing—original draft; Project administration; Writing—review and editing. **Marianne Becker**: Conceptualization; Investigation; Writing—review and editing. **Ulrike Schierloh**: Conceptualization; Investigation; Writing—review and editing. **Flávia Natividade da Silva**: Formal analysis; Investigation; Methodology; Writing—review and editing. **Leena Haataja**: Formal analysis; Investigation; Methodology; Writing—review and editing. **Ying Cai**: Investigation; Methodology. **Kashyap A Patel**: Data curation; Software; Formal analysis; Funding acquisition; Investigation; Methodology; Writing—review and editing. **Farah Kobaisi**: Formal analysis; Investigation; Methodology; Writing—review and editing. **Uyenlinh L Mirshahi**: Data curation; Formal analysis; Investigation; Methodology. **Kevin Colclough**: Investigation; Writing—review and editing. **Muhammad Shabab Javed**: Data curation; Software; Formal analysis; Investigation; Methodology; Writing—review and editing. **Matthew N Wakeling**: Data curation; Software; Formal analysis; Investigation; Methodology; Writing—review and editing. **Federica Fantuzzi**: Investigation; Writing—review and editing. **Maria Lytrivi**: Investigation; Methodology; Writing—review and editing. **Toshiaki Sawatani**: Investigation; Methodology; Writing—review and editing. **Maria Nicol Arroyo**: Software; Investigation; Methodology; Writing—review and editing. **Xiaoyan Yi**: Software; Investigation; Methodology; Writing—review and editing. **Chiara Vinci**: Investigation; Methodology. **Hossam Montaser**: Investigation; Methodology. **Nathalie Pachera**: Resources; Investigation; Methodology. **Timo Otonkoski**: Resources; Methodology. **Mariana Igoillo-Esteve**: Resources; Methodology.

**Raphaël Scharfmann**: Resources. **Andrew T Hattersley**: Resources; Funding acquisition. **Peter Arvan**: Conceptualization; Resources; Funding acquisition. **Carine De Beaufort**: Conceptualization; Resources. **Miriam Cnop**: Conceptualization; Resources; Formal analysis; Supervision; Funding acquisition; Project administration; Writing—review and editing.

Source data underlying figure panels in this paper may have individual authorship assigned. Where available, figure panel/source data authorship is listed in the following database record: biostudies:S-SCDT-10_1038-S44321-025-00362-9.

## Disclosure and competing interests statement

The authors declare no competing interests.

# Expanded View Figures

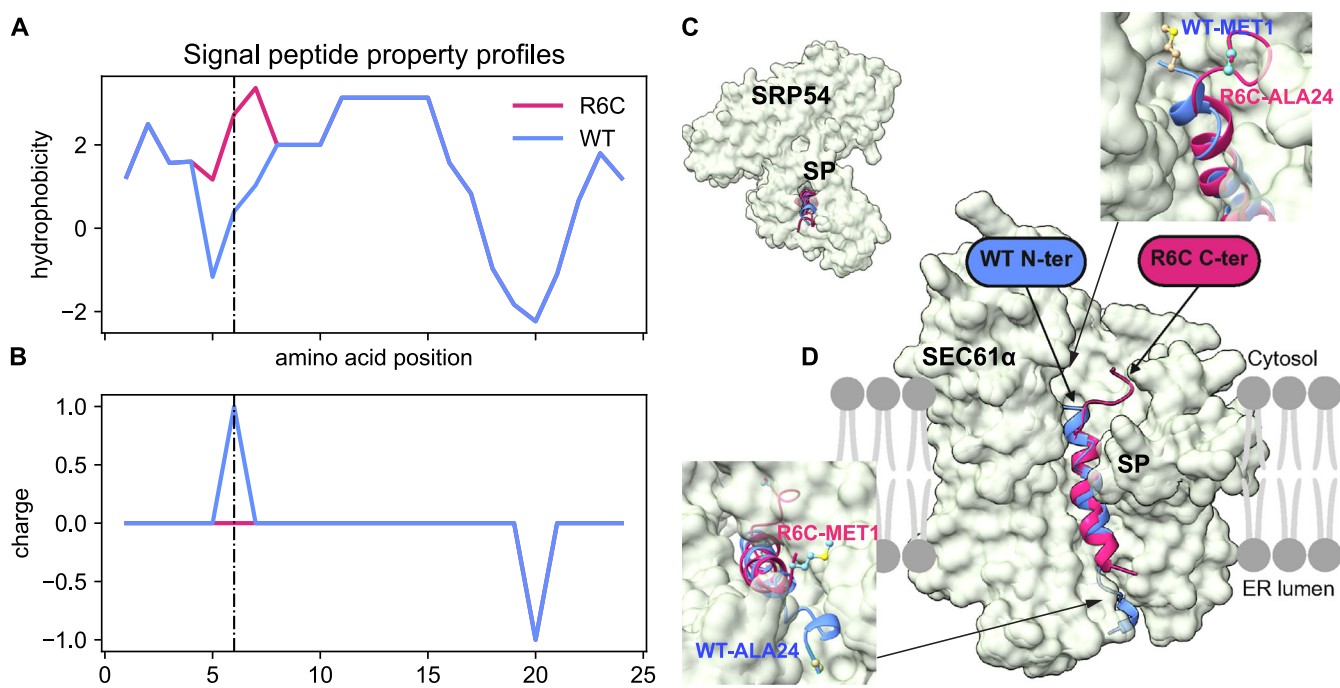

**Figure EV1. Comparison of wild-type and mutant signal peptide properties and interactions with SRP54 and SEC61 complex.**

Plotting and modeling used the first 24 amino acids of wildtype (WT, blue) or R6C (pink) preproinsulin. Hydrophobicity (**A**) and charge (**B**) profiles of the signal peptide (SP) for WT and R6C sequences. The dashed line indicates the position of the R6C substitution. (**C**) Structural model of aligned WT and R6C signal peptide bound to the SRP54 subunit, showing preserved orientation and overall binding. (**D**) Structural model of aligned WT and R6C signal peptide inserted into the SEC61α translocon, following an opposite insertion path into the ER membrane. Popped out window shows zoomed-in structure of cytosolic and lumen side of the ER, highlighting the first (1) and last (24) amino acids of either WT or R6C SP. Arabic number indicates the number of amino acids within the SP WT or R6C; MET methionine, ALA alanine. Scoring of both model details are listed in Appendix Table S4. Source data are available online for this figure.

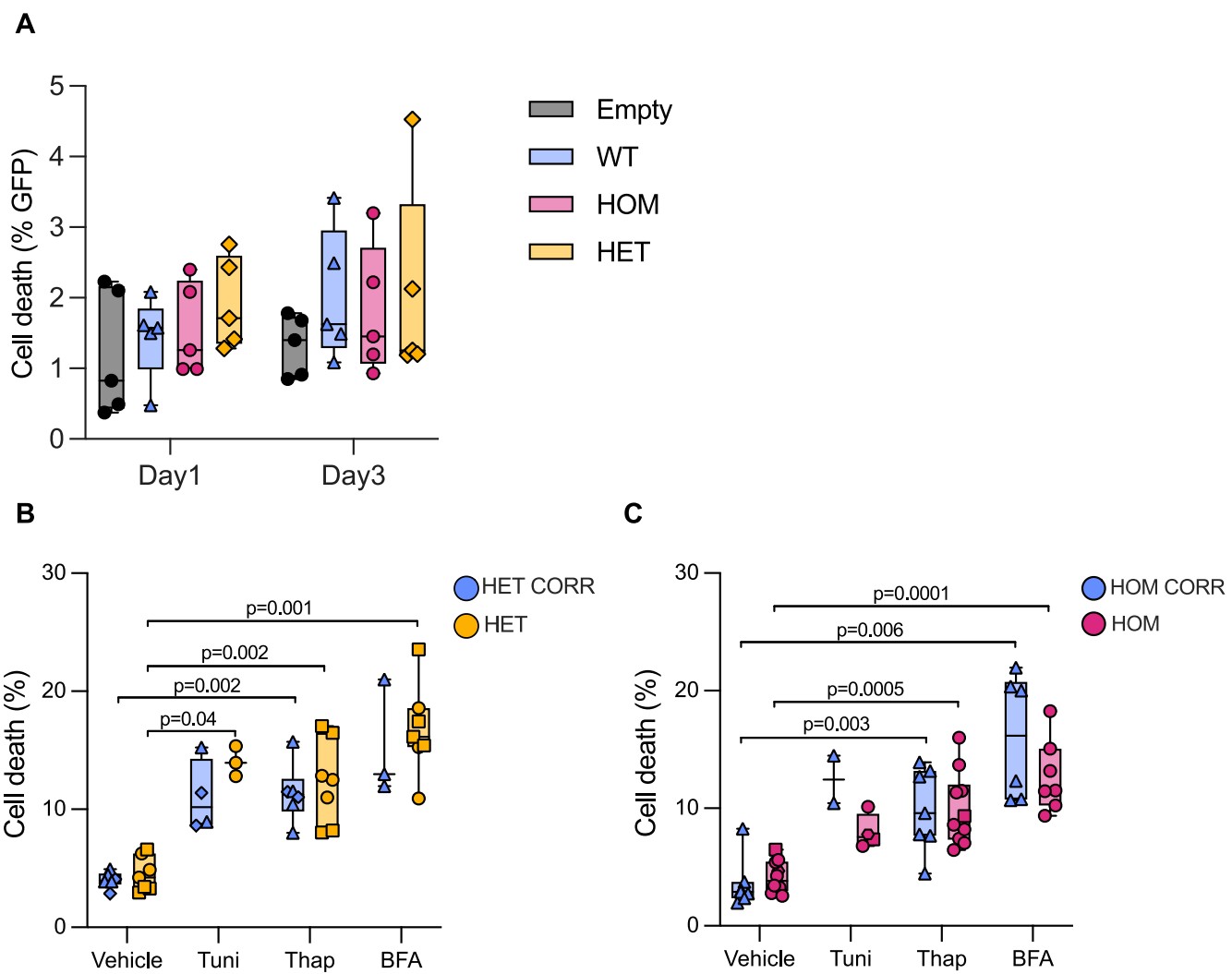

**Figure EV2.** *INS* R6C does not induce β-cell death.

(A) EndoC-βH1 INS-knockout cells were transfected with plasmids expressing wild-type (blue, WT) insulin, 100% R6C insulin (pink, R6C), 50% R6C + 50% wild-type insulin (yellow, HET), or GFP only (black, Empty) for 1 or 3 days. Cells were stained with propidium iodide (PI) and sorted for GFP expression and PI staining. Quantification of β cell death (%) as measured by percentage of double positivity for GFP and PI in total GFP positivity, $n = 5$. (B) Heterozygous R6C (HET, yellow) and isogenic corrected (HET CORR, blue) iPSCs and (C) homozygous (HOM, pink) and isogenic corrected (HOM CORR, blue) stage 7 iPSC-islets were exposed to synthetic ER stressors (Brefeldin A: BFA, 0.025 μg/mL, 24 h, thapsigargin: Thap, 1 μM, 48 h, or tunicamycin: Tuni, 5 μg/mL, 48 h) and cell death was assessed (%). Sample sizes (*n*) for each condition (HET CORR vs. HET; HOM CORR vs. HOM) were: Vehicle (DMSO), 6 vs. 7 and 7 vs. 10; BFA, 3 vs. 7 and 6 vs. 7; Thap, 6 vs. 7 and 7 vs. 10; Tuni, 4 vs. 3 and 2 vs. 4. Mixed-effects analysis with Tukey correction for multiple comparisons. In box plots, the median of independent experiments is shown by a horizontal line; 25th and 75th percentiles are at the bottom and top of the boxes; whiskers represent the minimum and maximum values. Source data are available online for this figure.

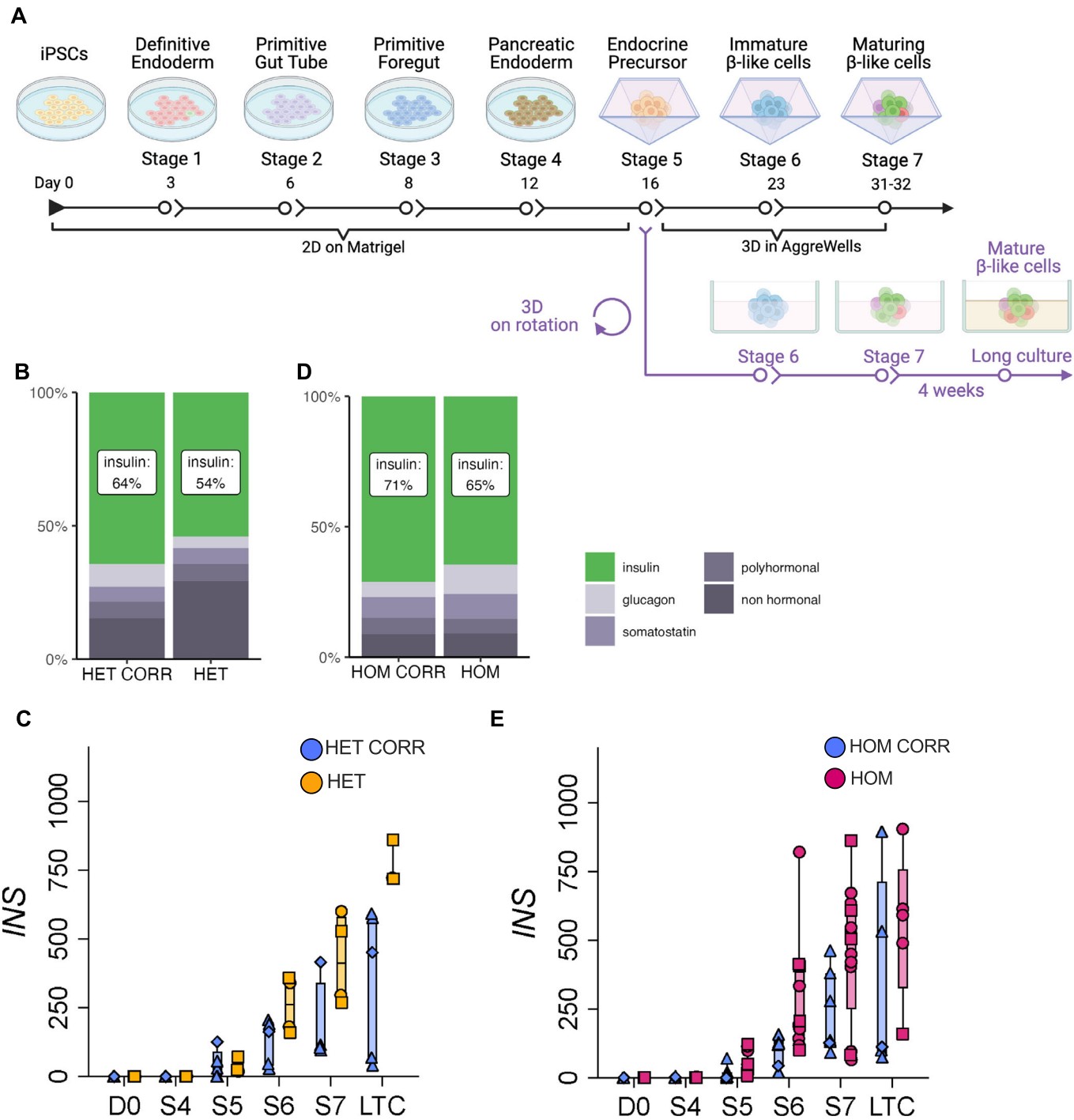

**Figure EV3.  R6C and corrected iPSCs differentiate into β cells.**

(A) Scheme illustrating the β cell differentiation protocol. (B) Heterozygous and corrected iPSC-islet cell composition as stained for insulin, glucagon, and somatostatin. HET CORR $n = 6$, HET $n = 7$. (C) *INS* mRNA expression in heterozygous R6C and corrected iPSC lines at iPSC stage (D0) and along differentiation stages to long-term culture (LTC). Data were normalized to the geometric mean of reference genes β-Actin and VAPA. HET CORR $n = 5$, HET $n = 4$. (D) Homozygous and corrected iPSC-islet cell composition as stained for insulin, glucagon, and somatostatin. HOM CORR $n = 6$, HOM $n = 14$. (E) *INS* mRNA expression in homozygous R6C and corrected iPSC lines at D0 and along differentiation stages to LTC. Data were normalized to the geometric mean of reference genes β-Actin and VAPA. HOM CORR $n = 8$, HOM $n = 15$. In box plots, the median of independent experiments is shown by a horizontal line; $25^{th}$ and $75^{th}$ percentiles are at the bottom and top of the boxes; whiskers represent the minimum and maximum values. Source data are available online for this figure.

