## [Peer Review File · EMBO Molecular Medicine]

A New Form of Diabetes Caused by INS Mutations Defined by Zygoty, Stem Cell and Population Data

Yue Tong, Marianne Becker, Ulrike Schierloh, Flávia Natividade da Silva, Leena Haataja, Ying Cai, Kashyap Patel, Farah Kobasi, Uyenlinh Mirshahi, Kevin Colclough, Muhammad Javed, Matthew Wakeling, Federica Fantuzzi, Maria Lytrivi, Toshiaki Sawatani, Maria Arroyo, Xiaoyan Yi, Chiara Vinci, Hossam Montaser, Nathalie Pachera, Timo Otonkoski, Mariana Igoillo-Esteve, Raphaël Scharfmann, Andrew Hattersley, Peter Arvan, Carine De Beaufort, and Miriam Cnop

Corresponding author: *Miriam Cnop* (miriam.cnop@ulb.be) , *Yue Tong* (yue.tong@ulb.be)

Review Timeline:

Submission Date:	11th Jul 25
Editorial Decision:	14th Aug 25
Revision Received:	14th Nov 25
Editorial Decision:	25th Nov 25
Revision Received:	27th Nov 25
Accepted:	28th Nov 25

Editor: Zeljko Durdevic

Transaction Report:

14th Aug 2025

Dear Prof. Cnop,

Thank you for the submission of your manuscript to EMBO Molecular Medicine and please accept my apologies for the delay in getting back to you due to the holiday season. We have now received feedback from the three reviewers who agreed to evaluate your manuscript. As you will see from the reports, all three referees are overall supportive of the study but also raise important concerns that should be addressed in a major revision. If you would like to discuss further the points raised by the referees, I am available to do so via email or video. Let me know if you are interested in this option.

We would welcome the submission of a revised version within three months for further consideration. Please let us know if you require longer to complete the revision.

I look forward to receiving your revised manuscript.

Yours sincerely,

Zeljko Durdevic

Zeljko Durdevic
Senior Editor
EMBO Molecular Medicine

We require:

- 1) A .docx formatted version of the manuscript text (including legends for main figures, EV figures and tables). Please make sure that the changes are highlighted to be clearly visible.
- 2) Individual production quality figure files as .eps, .tif, .jpg (one file per figure). For guidance, download the 'Figure Guide PDF': (<https://www.embopress.org/page/journal/17574684/authorguide#figureformat>).
- 3) A .docx formatted letter INCLUDING the reviewers' reports and your detailed point-by-point responses to their comments. As part of the EMBO Press transparent editorial process, the point-by-point response is part of the Review Process File (RPF), which will be published alongside your paper.
- 4) A complete author checklist, which you can download from our author guidelines (<https://www.embopress.org/page/journal/17574684/authorguide#submissionofrevisions>). Please insert information in the checklist that is also reflected in the manuscript. The completed author checklist will also be part of the RPF.
- 5) Please note that all corresponding authors are required to supply an ORCID ID for their name upon submission of a revised manuscript.
- 6) It is mandatory to include a 'Data Availability' section after the Materials and Methods. Before submitting your revision, primary

datasets produced in this study need to be deposited in an appropriate public database, and the accession numbers and database listed under 'Data Availability'. Please remember to provide a reviewer password if the datasets are not yet public (see <https://www.embopress.org/page/journal/17574684/authorguide#dataavailability>).

12) Author contributions: You will be asked to provide CRediT (Contributor Role Taxonomy) terms in the submission system. These replace a narrative author contribution section in the manuscript.

13) A Conflict of Interest statement should be provided in the main text.

14) Every published paper now includes a 'Synopsis' to further enhance discoverability. Synopses are displayed on the journal webpage and are freely accessible to all readers. They include a short stand first (maximum of 300 characters, including space) as well as 2-5 one-sentences bullet points that summarizes the paper. Please write the bullet points to summarize the key NEW findings. They should be designed to be complementary to the abstract - i.e. not repeat the same text. We encourage inclusion of key acronyms and quantitative information (maximum of 30 words / bullet point). Please use the passive voice. Please attach

these in a separate file or send them by email, we will incorporate them accordingly.

15) Include a Reagents and Tools Table as part of the Methods section, which can be downloaded from our author guidelines (<https://www.embopress.org/page/journal/17574684/authorguide#structuredmethods>)

***** Reviewer's comments *****

Referee #1 (Comments on Novelty/Model System for Author):

The approaches and models used are state of the art.

Referee #1 (Remarks for Author):

Review of Tong et al

General comments:

The manuscript from Tong and co-workers uses multiple approaches to assess the consequences of heterozygosity for the Ayg6Cys variant in the INS gene, which has been previously reported and is located in the signal peptide of preproinsulin. This is an important and timely paper that has ramifications for our understanding of rare types of diabetes, but also for the cell biology of the human beta-cell.

The authors use population-level analysis, as well as detailed studies of iPSC-derived beta-like cells, to provide evidence that this variant is a recessive pathogenic mutation. The authors largely rule out significant effects of the variant in the heterozygous state on proinsulin levels, insulin content, and insulin secretion.

Overall, this manuscript is very well written with clearly presented and convincing data. Once published, it will make an important contribution to the field.

This reviewer believes the manuscript is best published as a full-length article, rather than a short report.

Specific comments:

Cell death susceptibility, which is an important hard cellular endpoint, is measured in the EndoC-betaH1 cell line transfected with variant INS, wildtype, or a mix of the two. This is an interesting experiment, but it was not easy to interpret due to the low transfection rate and seemingly low n's. For most of the rest of the study, the authors move to iPSC-derived cells. However, they do not return to cell death or cell death susceptibility in a comprehensive way. Ideally, the authors would study cell death susceptibility of both the heterozygous and homozygous cells, and their controls, using a larger panel of culture conditions, including mild stresses. Also, the data in Fig S14 might be better shown side by side in the same graphs to compare the HOM CORR vs HOM.

The inset in Figure 3D doesn't seem to add much. The Ex4 response peaks at the same level, but starts much lower... is the correct interpretation that the HET CORR cells have a greater response? Should the AUC be corrected for the new 16.8 mM glucose baseline?

The title is a little unwieldy. This reviewer doesn't think it needs to list the types of data.

Referee #2 (Comments on Novelty/Model System for Author):

The manuscript titled "A New Form of Diabetes Caused by INS Mutations Defined by Zygosity Stem Cell and Population Data" by Yue Tong et al., provides a multidimensional evaluation of the INS R6C (p.Arg6Cys) insulin gene variant, integrating clinical genetics, population data, structural modelling, and functional analysis in iPSC-derived β cells and in vivo models. The major and novel claim is that the R6C variant is not a classical autosomal dominant mutation (as previously considered) but rather acts as a recessive (dose-dependent) pathogenic allele: homozygotes develop early-onset diabetes, while heterozygotes show variable or no phenotype. Overall, the manuscript makes an impactful contribution to the fields of monogenic diabetes and variant interpretation, offering a robust framework for classifying rare INS variants. The evidence supporting the reclassification of R6C as a recessive pathogenic variant is strong, drawing from well-integrated genetic, functional, and population data. Evaluating the effect on heterozygotes, supported by population evidence from large databases, is particularly important. These findings have meaningful implications for clinical management and the interpretation of uncertain INS variants. The technical quality appears high and methodologies are well described. Statistical analysis were also appropriate. The iPSC- β cell system is appropriate and well-validated for modeling patient-specific effects.

Referee #2 (Remarks for Author):

The manuscript titled "A New Form of Diabetes Caused by INS Mutations Defined by Zygosity Stem Cell and Population Data" by Yue Tong et al., provides a multidimensional evaluation of the INS R6C (p.Arg6Cys) insulin gene variant, integrating clinical genetics, population data, structural modelling, and functional analysis in iPSC-derived β cells and in vivo models. The major and novel claim is that the R6C variant is not a classical autosomal dominant mutation (as previously considered) but rather acts as a recessive (dose-dependent) pathogenic allele: homozygotes develop early-onset diabetes, while heterozygotes show variable or no phenotype. Functional studies in patient-derived iPSC lines powerfully recapitulate this gene dosage effect and provide mechanistic insight. However, additional data is required for long-term progression and penetrance in heterozygotes. Overall, the manuscript makes an impactful contribution to the fields of monogenic diabetes and variant interpretation, offering a robust framework for classifying rare INS variants. The evidence supporting the reclassification of R6C as a recessive pathogenic variant is strong, drawing from well-integrated genetic, functional, and population data. Evaluating the effect on heterozygotes, supported by population evidence from large databases, is particularly important. These findings have meaningful implications for clinical management and the interpretation of uncertain INS variants.

Major Comments

1. While the evidence for recessive nature is convincing, heterozygotes exhibit clinical heterogeneity, and some may have a mild phenotype or increased diabetes risk. The manuscript concludes that heterozygous R6C is not associated with increased diabetes risk, but this assertion is based on a limited sample size and short follow-up. The possibility of late-onset or environmentally triggered diabetes should be discussed more, as the current study may underestimate heterozygote penetrance.
2. The results section, while thorough, could be better organized. Key findings are sometimes buried in long paragraphs.
3. The methods section and introduction could be significantly condensed. Some experimental details may be moved to supplementary materials.
4. A more explicit discussion of metabolic, environmental, and genetic modifiers of monogenic diabetes risk would enrich the discussion and help contextualize heterozygote variability.

Minor Comments

1. Rewrite Line 184 to "(7% [6/86] vs. 6.7% [33,014/490,029]; Fisher's exact test, $P = 0.83$)" for improved clarity.
2. Rewrite Line 147 to improve sentence clarity.
3. On Line 74, better to list Wang et al, 2020 and Wang et al, 1999 separately.
4. Ensure consistent formatting: INS (italicized) for the gene, insulin (non-italicized) for the protein (e.g., Lines 170, 282, 364, etc.).
5. Two table headers are provided for Table S8.

Overall, an interesting study presenting an exemplary model for rare variant interpretation, integrating patient-derived iPSCs and population genomics to redefine disease mechanism and inheritance. Such a detailed study on a rare mutation is commendable.

Referee #3 (Remarks for Author):

This paper unambiguously demonstrates that the INS R6C variant causes diabetes only in the homozygous state, and that heterozygous carriers are not enriched in diabetes. This is an important and clinically useful finding. I have no concerns about the validity of the findings, but the paper can be much improved in a number of ways. Most importantly, as I explain below, the pathophysiology of dominant vs. recessive beta cell failure should be better explained as a justification of the experimental approach. Also, I find that the emphasis on methodological innovation is utterly misplaced, as detailed below.

1. The problem that this work solves should be better explained in the Introduction section, starting with a review of the clinical literature, going well beyond the first report by Edghill et al, 2008. The purpose of the Introduction section of a paper is to summarize what was known before the work was undertaken, and what question led to the study. It is not exactly a huge literature to review that can be easily searched by the NCBI LitVar2 feature.
2. The pathophysiology difference between the two genetic models (AD vs AR) should be better highlighted. Although most MODY forms involve haploinsufficiency, this is not likely to apply to INS, given the ability of the beta cell to modulate expression in response to feedback. Hyperproinsulinemia (OMIM #616214) is a good example and, indeed, the literature cited clearly shows that pathogenic heterozygous variants always involve damage to the beta cell by misfolded peptide. This distinction should be made more clearly in the Abstract as well.
3. The in vitro work is convincing enough, but much stronger evidence comes from a much newer methodological innovation, the ability to search the ever increasingly comprehensive population databases based on genotype rather than phenotype. Work with iPSC-generated beta cells is a powerful tool, but it has become routine over the past decade. Language like "This study highlights the power of patients' iPSC-based disease modeling" should be toned down, while the ability to clearly define pathogenicity from population databases should be clearly defined as the methodological innovation.
4. The mouse transplantation experiment adds little to the in vitro results and should be shortened if not completely eliminated.
5. The paper is much too long for what it has to say, and can be drastically shortened. Trivial routine methodologies, such as Sanger sequencing and Western blot need not be described in any detail. The Discussion is unnecessarily wordy and can be cut down to about half its current size focusing on what is both new and interesting.
6. The Results section should start with phenotype, not genotype. Describe how the proband was ascertained, her clinical features, and what led to genetic exploration (even if this is already obvious by looking at the pedigree), before giving sequencing results.
7. Lines 178 and 184 give two drastically different numbers for the heterozygotes in UKB.
8. Please check the math in line 183. It doesn't add up

Minor suggested corrections

1. Line 44. Did you mean "post-transcriptionally?"
2. The sentence "This indicates. . . mutation" in Line 138 is totally redundant, as it repeats what was stated in the previous sentence in different words.
3. For GRS2, reference to ten SNPs is meaningless, as different variants have drastically different betas. Just state the score percentile.
4. Line 186. As a dominant, the variant is clearly benign according to recently proposed criteria (PMID 39379762, PMID 39402157) and should be referred to as such.

Authors' own comments

While the paper was under review, we identified another patient with childhood-onset diabetes caused by a homozygous *INS* R6C variant. This proband was diagnosed with diabetes at 13 years with HbA1c 78 mM/M (9.3%) and BMI 27.2 kg/m². These new data have been included in the revised manuscript (Figure 1B, Table 1), expanding the clinical findings. For the data availability section, reviewer's token is sbktnmgybnarxkv.

Referee #1 (Comments on Novelty/Model System for Author):

The approaches and models used are state of the art.

Referee #1 (Remarks for Author):

Review of Tong et al

General comments:

The manuscript from Tong and co-workers uses multiple approaches to assess the consequences of heterozygosity for the Ays6Cys variant in the *INS* gene, which has been previously reported and is located in the signal peptide of proinsulin. This is an important and timely paper that has ramifications for our understanding of rare types of diabetes, but also for the cell biology of the human beta-cell.

The authors use population-level analysis, as well as detailed studies of iPSC-derived beta-like cells, to provide evidence that this variant is a recessive pathogenic mutation. The authors largely rule out significant effects of the variant in the heterozygous state on proinsulin levels, insulin content, and insulin secretion.

Overall, this manuscript is very well written with clearly presented and convincing data. Once published, it will make an important contribution to the field.

This reviewer believes the manuscript is best published as a full-length article, rather than a short report.

We are grateful to the reviewer for the very positive comments and delighted that they found the study important, the approaches state of the art, and the data clearly presented and convincing.

Specific comments:

Cell death susceptibility, which is an important hard cellular endpoint, is measured in the EndoC-betaH1 cell line transfected with variant *INS*, wildtype, or a mix of the two. This is an interesting experiment, but it was not easy to interpret due to the low transfection rate and seemingly low n's. For most of the rest of the study, the authors move to iPSC-derived cells. However, they do not return to cell death or cell death susceptibility in a comprehensive way. Ideally, the authors would study cell death susceptibility of both the heterozygous and homozygous cells, and their controls, using a larger panel of culture conditions, including mild stresses. Also, the data in Fig S14 might be better shown side by side in the same graphs to compare the HOM CORR vs HOM.

We thank the reviewer for the valid comments. We increased the number of experiments in EndoC- β H1 cells to $n=5$ (Figure EV2A and Appendix Figure S2). The transfection rate reaches up to 35-40% after 3 days (Appendix Figure S2C), which provides sufficient transfected cell numbers (Appendix Figure S2A) to reliably assess cell death events. In this experimental model, few dead cells were detected (around 0.5% of all cells (Appendix Figure S2B) and around 1.5% of the transfected cells after 3 days (Figure EV2A)) and there was no difference between cells transfected with wildtype, heterozygous or homozygous R6C INS (Figure EV2A). These data are described in the results section on page 8.

We have now further studied cell death susceptibility in homozygous and heterozygous INS R6C iPSC-derived β cells and their isogenic controls using a panel of culture conditions. We had previously examined cell death in homozygous iPSC-derived β cells exposed to thapsigargin and brefeldin A (old Figure S14A). We have now included experiments in which we exposed heterozygous iPSC- β cells to 3 different ER stressors, namely tunicamycin, thapsigargin and brefeldin A, and we also exposed homozygous iPSC- β cells to tunicamycin, an ER stressor that is milder than thapsigargin and brefeldin A. These new cell death data have been included in the new Figure EV2B and C, and they are described in the results section on page 13. To facilitate the comparison of HOM CORR vs HOM, we now show the data of Fig S14F-I side by side in the same graph (Appendix Figure S17).

The inset in Figure 3D doesn't seem to add much. The Ex4 response peaks at the same level, but starts much lower... is the correct interpretation that the HET CORR cells have a greater response? Should the AUC be corrected for the new 16.8 mM glucose baseline?

Following the reviewer's suggestion, the insert of Figure 3D has been removed. The exendin-4 stimulation peaks at the same level for HET and HET CORR cells, indeed. In relative terms this might suggest a greater stimulation by the GLP-1 receptor agonist in HET CORR cells, but in absolute terms there is no difference (see also Figure 3G). The AUC panels (Figure 3E-H) show absolute insulin output, not a relative value or stimulation index corrected for baseline insulin secretion. We prefer to keep this presentation because it shows actual data rather than fold changes.

The title is a little unwieldy. This reviewer doesn't think it needs to list the types of data.

We thank the reviewer for this comment and would appreciate editorial advice on whether the title should be modified. For the time being, we have kept the title as it is, to put forward that zygosity is important for the INS R6C variant clinical phenotype, and that this is supported by stem cell experiments as well as population data.

Referee #2 (Comments on Novelty/Model System for Author):

The manuscript titled "A New Form of Diabetes Caused by INS Mutations Defined by Zygosity Stem Cell and Population Data" by Yue Tong et al., provides a multidimensional evaluation of the INS R6C (p.Arg6Cys) insulin gene variant, integrating clinical genetics, population data, structural modelling, and functional analysis in iPSC-derived β cells and in vivo models. The major and novel claim is that the R6C variant is not a classical autosomal dominant mutation (as previously considered) but rather acts as a recessive (dose-dependent) pathogenic allele:

homozygotes develop early-onset diabetes, while heterozygotes show variable or no phenotype. Overall, the manuscript makes an impactful contribution to the fields of monogenic diabetes and variant interpretation, offering a robust framework for classifying rare INS variants. The evidence supporting the reclassification of R6C as a recessive pathogenic variant is strong, drawing from well-integrated genetic, functional, and population data. Evaluating the effect on heterozygotes, supported by population evidence from large databases, is particularly important. These findings have meaningful implications for clinical management and the interpretation of uncertain INS variants.

The technical quality appears high and methodologies are well described. Statistical analysis were also appropriate. The iPSC- β cell system is appropriate and well-validated for modeling patient-specific effects.

Referee #2 (Remarks for Author):

The manuscript titled "A New Form of Diabetes Caused by INS Mutations Defined by Zygosity Stem Cell and Population Data" by Yue Tong et al., provides a multidimensional evaluation of the INS R6C (p.Arg6Cys) insulin gene variant, integrating clinical genetics, population data, structural modelling, and functional analysis in iPSC-derived β cells and in vivo models. The major and novel claim is that the R6C variant is not a classical autosomal dominant mutation (as previously considered) but rather acts as a recessive (dose-dependent) pathogenic allele: homozygotes develop early-onset diabetes, while heterozygotes show variable or no phenotype. Functional studies in patient-derived iPSC lines powerfully recapitulate this gene dosage effect and provide mechanistic insight. However, additional data is required for long-term progression and penetrance in heterozygotes.

Overall, the manuscript makes an impactful contribution to the fields of monogenic diabetes and variant interpretation, offering a robust framework for classifying rare INS variants. The evidence supporting the reclassification of R6C as a recessive pathogenic variant is strong, drawing from well-integrated genetic, functional, and population data. Evaluating the effect on heterozygotes, supported by population evidence from large databases, is particularly important. These findings have meaningful implications for clinical management and the interpretation of uncertain INS variants.

We are grateful to the reviewer for the very positive comments on our multidimensional evaluation of the *INS* R6C variant and delighted that they found the technical quality high and the findings impactful.

Major Comments

1. While the evidence for recessive nature is convincing, heterozygotes exhibit clinical heterogeneity, and some may have a mild phenotype or increased diabetes risk. The manuscript concludes that heterozygous R6C is not associated with increased diabetes risk, but this assertion is based on a limited sample size and short follow-up. The possibility of late-onset or environmentally triggered diabetes should be discussed more, as the current study may underestimate heterozygote penetrance.

We thank the reviewer for the valid comment. The number of reported individuals carrying a heterozygous *INS* R6C variant is indeed limited, and insufficient to determine penetrance. The large-scale population data provide evidence on higher numbers of heterozygous R6C carriers: in the UK Biobank (4 carriers) and in the Geisinger MyCode cohort (56 carriers) no increase

was seen in diabetes risk as compared to the general population. It remains possible that R6C diabetes manifests later in life or is favored by environmental triggers. We now discuss this on pages 16-17.

2. The results section, while thorough, could be better organized. Key findings are sometimes buried in long paragraphs.

The comment is well taken. We have better organized the results section and increased brevity.

3. The methods section and introduction could be significantly condensed. Some experimental details may be moved to supplementary materials.

We thank the reviewer for this suggestion. We have tried to condense the introduction, while, in response to the request by reviewer 3, reviewing the literature and outlining the questions addressed by the study. Experimental details have been moved to the Appendix.

4. A more explicit discussion of metabolic, environmental, and genetic modifiers of monogenic diabetes risk would enrich the discussion and help contextualize heterozygote variability.

We thank the reviewer for this suggestion. We have now developed an explicit discussion on how genetic modifiers and/or metabolic and environmental factors contribute to monogenic diabetes risk on pages 16-17.

Minor Comments

1. Rewrite Line 184 to "(7% [6/86] vs. 6.7% [33,014/490,029]; Fisher's exact test, $P = 0.83$)" for improved clarity.

Thank you for the suggestion; this has now been clarified on page 5.

2. Rewrite Line 147 to improve sentence clarity.

This is now better explained in Table 1 (page 41). This individual stopped insulin treatment at 18 years on his own initiative, because of poorly controlled glycemia.

3. On Line 74, better to list Wang et al, 2020 and Wang et al, 1999 separately.

This has been done (page 3).

4. Ensure consistent formatting: INS (italicized) for the gene, insulin (non-italicized) for the protein (e.g., Lines 170, 282, 364, etc.).

Thank you for the suggestion; this has been done throughout the manuscript.

5. Two table headers are provided for Table S8.

The extra header has been removed.

Overall, an interesting study presenting an exemplary model for rare variant interpretation, integrating patient-derived iPSCs and population genomics to redefine disease mechanism and inheritance. Such a detailed study on a rare mutation is commendable.

Referee #3 (Remarks for Author):

This paper unambiguously demonstrates that the *INS* R6C variant causes diabetes only in the homozygous state, and that heterozygous carriers are not enriched in diabetes. This is an important and clinically useful finding. I have no concerns about the validity of the findings, but the paper can be much improved in a number of ways. Most importantly, as I explain below, the pathophysiology of dominant vs. recessive beta cell failure should be better explained as a justification of the experimental approach. Also, I find that the emphasis on methodological innovation is utterly misplaced, as detailed below.

We are grateful to the reviewer for the positive comments. We are delighted that they found the study important and clinically useful, and the findings valid. The suggestions for improvement are very well taken, and the manuscript has been revised accordingly.

1. The problem that this work solves should be better explained in the Introduction section, starting with a review of the clinical literature, going well beyond the first report by Edghill et al, 2008. The purpose of the Introduction section of a paper is to summarize what was known before the work was undertaken, and what question led to the study. It is not exactly a huge literature to review that can be easily searched by the NCBI LitVar2 feature.

We thank the reviewer for this excellent suggestion. The introduction has been updated with a brief review (page 5) on what is known and unknown about the *INS* R6C mutations after the first report, both in terms of mechanistic confirmation studies and in clinical investigations.

2. The pathophysiology difference between the two genetic models (AD vs AR) should be better highlighted. Although most *MODY* forms involve haploinsufficiency, this is not likely to apply to *INS*, given the ability of the beta cell to modulate expression in response to feedback. Hyperproinsulinemia (OMIM #616214) is a good example and, indeed, the literature cited clearly shows that pathogenic heterozygous variants always involve damage to the beta cell by misfolded peptide. This distinction should be made more clearly in the Abstract as well.

We thank the reviewer for this valid comment. We have highlighted the important concept of different pathophysiology in dominant vs recessive models in the abstract as well as in the introduction (page 3).

3. The in vitro work is convincing enough, but much stronger evidence comes from a much newer methodological innovation, the ability to search the ever increasingly comprehensive population databases based on genotype rather than phenotype. Work with iPSC-generated beta cells is a powerful tool, but it has become routine over the past decade. Language like "This study highlights the power of patients' iPSC-based disease modeling" should be toned down, while the ability to clearly define pathogenicity from population databases should be clearly defined as the methodological innovation.

Following the reviewer's suggestion, we have toned down the claims related to iPSC-based disease modeling (abstract page 2). We emphasize the importance of population database-derived evidence to define pathogenicity (abstract page 2, and discussion page 15).

4. The mouse transplantation experiment adds little to the in vitro results and should be shortened if not completely eliminated.

Following the reviewer's suggestion, we have significantly shortened the mouse transplantation experiment section and moved some of the data to the Appendix (Figure 6 and Appendix Figure S13, results pages 11-12).

5. The paper is much too long for what it has to say, and can be drastically shortened. Trivial routine methodologies, such as Sanger sequencing and Western blot need not be described in any detail. The Discussion is unnecessarily wordy and can be cut down to about half its current size focusing on what is both new and interesting.

We thank the reviewer for these suggestions. We have shortened the methods section, moving some experimental details to the Appendix (also following the request of reviewer 2). The discussion section has been shortened.

6. The Results section should start with phenotype, not genotype. Describe how the proband was ascertained, her clinical features, and what led to genetic exploration (even if this is already obvious by looking at the pedigree), before giving sequencing results.

We thank the reviewer for this excellent comment. We now describe the phenotype of the proband and her uncle, both of whom developed diabetes in childhood. While the paper was under review, we identified another patient with childhood-onset diabetes caused by a homozygous *INS* R6C variant. This proband's phenotype has also been included. The results of the genetic analyses are now provided after the description of the clinical features (pages 5-6).

7. Lines 178 and 184 give two drastically different numbers for the heterozygotes in UKB.

The different numbers given for the prevalence of heterozygotes in the UK Biobank were for *INS* R6C and *INS* R6H variants. The text has been edited to clearly indicate this (page 7).

8. Please check the math in line 183. It doesn't add up

Thank you for pointing this out. We have clarified the text to distinguish allele frequency from heterozygote frequency, which led to the apparent math problem (page 7).

Minor suggested corrections

1. Line 44. Did you mean "post-transcriptionally?"

We meant at the level of transcription (RNA-seq, mRNA). We have now clarified this in the abstract (page 2) as well as in the discussion (page 15).

2. The sentence "This indicates. . . mutation" in Line 138 is totally redundant, as it repeats what was stated in the previous sentence in different words.

The comment is well taken; the repetitive sentence has been removed (page 6).

3. For GRS2, reference to ten SNPs is meaningless, as different variants have drastically different betas. Just state the score percentile.

Thank you for the valid comment; reference to the SNPs has been removed (page 6).

4. Line 186. As a dominant, the variant is clearly benign according to recently proposed criteria (PMID 39379762, PMID 39402157) and should be referred to as such.

We thank the reviewer for this excellent comment. We have edited the text to clearly state this and provide the references (page 7).

25th Nov 2025

Dear Prof. Cnop,

Thank you for the submission of your revised manuscript to EMBO Molecular Medicine. I am pleased to inform you that we will be able to accept your manuscript pending the following final amendments:

1) In the main manuscript file, please do the following:

- Please address all comments suggested by our data editors listed below:

o Figure legends:

1. Please note that the legends for figure EV 3 are not provided in the sequential manner. This needs to be rectified.

2. Please indicate the statistical test used for data analysis in the legends of figures 6b, d; 7d; EV 2b, c.

3. Please note that the box plots need to be defined in terms of minima, maxima, centre, bounds of box and whiskers, and percentile in the legends of figures 7b, d.

- Add up to 5 keywords.

- Correct callouts for Table S4 to Appendix Table S4.

- In Methods, please provide the statement that informed consent was obtained from all human subjects and that the experiments conformed to the principles set out in the WMA Declaration of Helsinki and the Department of Health and Human Services Belmont Report.

- Remove BioRender reference from legends and add a dedicated section to the Methods:

Graphics:

(some of the... OR Figure #... OR synopsis) Graphics were created with BioRender.com.

- Indicate in legends exact n and exact p values, not a range, along with the statistical test used. To keep the figures "clear" some authors found providing an Appendix table Sx with all exact p-values preferable. You are welcome to do this if you want to.

- Author contributions: Please remove it from the manuscript and specify author contributions in our submission system. CRediT has replaced the traditional author contributions section because it offers a systematic machine-readable author contributions format that allows for more effective research assessment. You are encouraged to use the free text boxes beneath each contributing author's name to add specific details on the author's contribution. More information is available in our guide to authors:

<https://www.embopress.org/page/journal/17574684/authorguide#authorshipguidelines>

- In the expanded view figure legends, rename the figures to "Figure EV1" etc.

2) Appendix: Please move Appendix Methods to main Methods section. Also, rearrange figures so that the legends are placed underneath the corresponding figures.

3) Tables: Please upload the Appendix Table 7 as a Dataset in excel format, rename it to Dataset EV1 and update its callouts in the main text. Remaining Appendix tables should be renumbered and their callouts updated.

4) Synopsis:

- Synopsis image: Please provide the image as a high-resolution jpeg file 550 px-wide x 300-600 pixels high.

- Synopsis text: Please remove it from the manuscript and upload it as a separate .doc file.

5) As part of the EMBO Publications transparent editorial process (see our Editorial at

<http://embomolmed.embopress.org/content/2/9/329>), EMBO Molecular Medicine will publish online a Review Process File (RPF) to accompany accepted manuscripts. This file will be published in conjunction with your paper and will include the anonymous referee reports, your point-by-point response and all pertinent correspondence relating to the manuscript. Let us know whether you agree with the publication of the RPF and as here, if you want to remove or not any figures from it prior to publication.

6) Please provide a point-by-point letter INCLUDING my comments as well as the reviewer's reports and your detailed responses (as Word file).

I look forward to reading a new revised version of your manuscript as soon as possible.

Yours sincerely,

Zeljko Durdevic

Zeljko Durdevic
Senior Editor
EMBO Molecular Medicine

*** Instructions to submit your revised manuscript ***

When preparing your revised manuscript, please refer to our guidelines: <https://link.springer.com/journal/44321/submission-guidelines#cms-Revised-submissions>. We perform an initial quality control of all revised manuscripts before re-review; failure to include requested items will delay the evaluation of your revision.

We require:

2) Individual production quality figure files as .eps, .tif, .jpg (one file per figure). For guidance, download the 'Figure Guide PDF': <https://media.springernature.com/original/springer-cms/rest/v1/content/27825798/data/v1>.

3) A .docx formatted letter INCLUDING the reviewers' reports and your detailed point-by-point responses to their comments. As part of the EMBO Press transparent editorial process, the point-by-point response is part of the Review Process File (RPF), which will be published alongside your paper.

4) A complete author checklist, which you can download from our author guidelines. Please insert information in the checklist that is also reflected in the manuscript. The completed author checklist will also be part of the RPF.

6) It is mandatory to include a 'Data Availability' section after the Materials and Methods. Before submitting your revision, primary datasets produced in this study need to be deposited in an appropriate public database, and the accession numbers and database listed under 'Data Availability'. Please remember to provide a reviewer password if the datasets are not yet public.

7) For data quantification: please specify the name of the statistical test used to generate error bars and P values, the number (n) of independent experiments (specify technical or biological replicates) underlying each data point and the test used to calculate p-values in each figure legend. The figure legends should contain a basic description of n, P and the test applied. Graphs must include a description of the bars and the error bars (s.d., s.e.m.).

9) Our journal encourages inclusion of *data citations in the reference list* to directly cite datasets that were re-used and obtained from public databases. Data citations in the article text are distinct from normal bibliographical citations and should directly link to the database records from which the data can be accessed. In the main text, data citations are formatted as follows: "Data ref: Smith et al, 2001" or "Data ref: NCBI Sequence Read Archive PRJNA342805, 2017". In the Reference list, data citations must be labeled with "[DATASET]". A data reference must provide the database name, accession number/identifiers and a resolvable link to the landing page from which the data can be accessed at the end of the reference.

10) We replaced Supplementary Information with Expanded View (EV) Figures and Tables that are collapsible/expandable

online. A maximum of 5 EV Figures can be typeset. EV Figures should be cited as 'Figure EV1, Figure EV2' etc... in the text and their respective legends should be included in the main text after the legends of regular figures.

- the medical issue you are addressing,

- the results obtained and

- their clinical impact.

12) Author contributions: You will be asked to provide CRediT (Contributor Role Taxonomy) terms in the submission system. These replace a narrative author contribution section in the manuscript.

13) A Disclosure and competing interests statement should be provided in the main text.

14) Every published paper includes a 'Synopsis' to further enhance discoverability. Synopses are displayed on the journal webpage and are freely accessible to all readers. They include a short stand first (maximum of 300 characters, including space) as well as 2-5 one-sentences bullet points that summarizes the paper. Please write the bullet points to summarize the key NEW findings. They should be designed to be complementary to the abstract - i.e. not repeat the same text. We encourage inclusion of key acronyms and quantitative information (maximum of 30 words / bullet point). Please use the passive voice. Please attach these in a separate file or send them by email, we will incorporate them accordingly.

15) Include a Reagents and Tools Table as part of the Methods section, which can be downloaded from our author guidelines.

Photos 400-800 DPI

*Additional important information regarding figures and illustrations can be found at <https://media.springernature.com/original/springer-cms/rest/v1/content/27825798/data/v1>

***** Reviewer's comments *****

Referee #2 (Remarks for Author):

The revised manuscript is now stronger.

Referee #3 (Comments on Novelty/Model System for Author):

The paper uses state-of-the-art methodologies in a meaningful way. medical impact, is the only feature that I rated less than high, and this is solely because of the relative rarity of the clinical problem

Referee #3 (Remarks for Author):

The revised version now meets the high standards of the journal and it is suitable for publication. I have no further revisions to suggest.

The authors addressed the remaining editorial issues.

28th Nov 2025

Dear Prof. Cnop,

We are pleased to inform you that your manuscript is accepted for publication and is now being sent to our publisher to be included in the next available issue of EMBO Molecular Medicine.

Zeljko Durdevic
Senior Editor
EMBO Molecular Medicine
